# Slot-Guided Adaptation of Pre-trained Diffusion Models for Object-Centric Learning and Compositional Generation

**Adil Kaan Akan**[1]       **Yucel Yemez**[1,2]

[1]Department of Computer Engineering, Koc University  [2]KUIS AI Center, Koc University

## Abstract

We present SlotAdapt, an object-centric learning method that combines slot attention with pretrained diffusion models by introducing adapters for slot-based conditioning. Our method preserves the generative power of pretrained diffusion models, while avoiding their text-centric conditioning bias. We also incorporate an additional guidance loss into our architecture to align cross-attention from adapter layers with slot attention. This enhances the alignment of our model with the objects in the input image without using external supervision. Experimental results show that our method outperforms state-of-the-art techniques in object discovery and image generation tasks across multiple datasets, including those with real images. Furthermore, we demonstrate through experiments that our method performs remarkably well on complex real-world images for compositional generation, in contrast to other slot-based generative methods in the literature. The project page can be found at https://kaanakan.github.io/SlotAdapt/.

## 1 Introduction

The real world is inherently structured with distinct, composable parts and objects that can be combined in various ways; this compositional characteristic is essential for cognitive functions like reasoning, understanding causality, and ability to generalize beyond training data (Lake et al., 2017; Bottou, 2014; Schölkopf et al., 2021; Bahdanau et al., 2019; Fodor & Pylyshyn, 1988). While language clearly reflects this modularity through sentences made up of distinct words and tokens, the compositional structure is less obvious in visual data. Object-centric learning (OCL) offers a promising approach to uncover this latent structure by grouping related features into coherent object representations without supervision (Kahneman et al., 1992; Greff et al., 2020). By decomposing complex scenes into separate objects and their interactions, OCL mimics how humans interpret their environment (Spelke & Kinzler, 2007), potentially improving the robustness and interpretability of AI systems (Lake et al., 2017; Schölkopf et al., 2021). This approach shifts from traditional pixel-based feature extraction to a more meaningful segmentation of visual data, which is key for better generalization and supporting high-level reasoning tasks.

Recent advances in OCL have shown the potential to incorporate powerful generative models, such as transformers and diffusion models, into the OCL framework as image decoders. Notably, models such as Latent Slot Diffusion (LSD) (Jiang et al., 2023) and SlotDiffusion (Wu et al., 2023b) have considerably improved performance in object discovery and visual generation tasks in real-world settings by employing slot-conditioned diffusion models. A concurrent work, GLASS (Singh et al., 2024), has made further progress in handling complex natural scenes by using generated captions as external guiding signals.

There exist two main approaches of integrating diffusion models into slot-based OCL methods in order to deal with real-world images. The first one, as seen in GLASS and LSD, directly uses a pretrained stable diffusion model as decoder. While this approach attempts to fully harness the generative power of pretrained diffusion models, it relies on cross-attention layers to achieve slot conditioning, which were however optimized to receive text input. In contrast, the second approach, as in SlotDiffusion, trains the diffusion model from scratch on the target dataset, thereby eliminating any such text-conditioning related biases. However, this limits the generation capacity of the slot-conditioned model, making it less effective in handling complex real-world images.

Enabling slot-based methods to fully exploit the generative capabilities of pretrained diffusion models while avoiding biases due to text-trained conditioning is a major challenge in object-centric learning, which hinders the effective handling of complex real-world images, particularly in compositional image generation tasks (Jung et al., 2024). In this work, we propose a novel approach that addresses this challenge by combining slot attention (Locatello et al., 2020) with adaptive conditioning in diffusion models (Mou et al., 2024; Rombach et al., 2022; Sohl-Dickstein et al., 2015; Ho et al., 2020). Our method, that we refer to as SlotAdapt, introduces the use of adapter layers (Mou et al., 2024) for slot conditioning, which eliminates text-trained conditioning bias in pretrained diffusion models and allows the slots to diverge from representations in the text embedding space while retaining the generative power of the pretrained diffusion model.

Another challenge for OCL methods is dealing with the part-whole hierarchy problem – the difficulty of deciding whether to segment an object into its parts or as a whole (Hinton, 1979), especially in real-world settings. The GLASS model (Singh et al., 2024) mitigates this problem with the aid of external supervision and at the cost of increased complexity, whereas Jung et al. (2024) introduce compositional representation learning; however the applicability of their method on real-world datasets remains limited. In order to address this challenge, we propose using cross-attention masks from adapter layers as pseudo-ground truth to guide slot attention maps. This self-supervisory signal enhances the alignment between learned slots and actual objects without external supervision, facilitating the learning of meaningful object representations. In fact, in our architecture, the guidance is mutual, as the cross-attention masks in the diffusion model are also guided and refined jointly with the slot attention masks, improving the image generation process.

In summary, we have the following main contributions: 1) We introduce adapters for slot-based conditioning to combine slot attention with pretrained diffusion models, 2) We propose an additional guidance loss to align cross-attention from adapter layers with slot attention without using external supervision, and 3) We present compositional image generation results on complex real-world images.

Through extensive experiments on various datasets, we demonstrate that our method, SlotAdapt, outperforms previous approaches in object discovery and compositional image generation tasks (Jiang et al., 2023; Wu et al., 2023b; Seitzer et al., 2023), especially on complex real-world image datasets. The performance of SlotAdapt in segmentation task is similar with the concurrent work (Singh et al., 2024), yet we achieve this with significantly reduced computational requirements and without any external supervision. Note that GLASS does not address the compositional generation task. To the best of our knowledge, our work is the first to present successful results for compositional generation on COCO (Lin et al., 2014), a large complex real-world image dataset with 80 different object classes.

## 2 RELATED WORK

### 2.1 UNSUPERVISED OBJECT-CENTRIC LEARNING

Unsupervised object-centric learning aims to decompose visual scenes into meaningful object representations without need for annotation. Early methods used iterative inference and CNN decoders to reconstruct scenes from object-specific feature vectors (*slots*) (Eslami et al., 2016; Burgess et al., 2019; Greff et al., 2019; Lin et al., 2019; Jiang et al., 2019; Lin et al., 2020). Attend-Infer-Repeat (AIR) (Eslami et al., 2016) and Sequential AIR (SQAIR) (Kosiorek et al., 2018) reconstructed objects in canonical poses using patch-based decoders. Slot attention (Locatello et al., 2020) and SAVi (Kipf et al., 2021) employed the spatial broadcast decoder (Watters et al., 2019) to predict images and segmentation masks from slots, combined via alpha compositing. While effective on synthetic datasets like CLEVR (Johnson et al., 2017) and 3D Shapes (Burgess & Kim, 2018), these methods struggle with complex, real-world images. Transformer-based decoders, such as SLATE (Singh et al., 2021) and STEVE (Singh et al., 2022), pre-train a discrete VAE (dVAE) to tokenize images and use slot-conditioned transformers for autoregressive reconstruction. Despite improvements, they face challenges in the high-quality reconstruction of complex real images (Singh et al., 2021; Wu et al., 2023a). Methods like DINOSAUR (Seitzer et al., 2023) bypass reconstruction by using self-supervised learning to discover objects but lack image generation capabilities. To improve slot-based autoencoders, Kakogeorgiou et al. (2024) propose self-training and patch-order permutation strategies, enhancing segmentation in complex real-world images. A relatively recent research direction in OCL is to employ diffusion models as slot decoders, enhancing scene decomposition and visual fidelity in real-world scenarios (Jiang et al., 2023; Wu et al., 2023b). Several approaches

have tackled compositional scene generation through different mechanisms: combining global and symbolic latent variables (Jiang & Ahn, 2020), using hierarchical VAEs with slot attention (Wang et al., 2023), and leveraging hierarchical discrete representations with autoregressive transformers (Wu et al., 2024). While these works focus on synthetic datasets, our work can achieve compositional generation in real-world scenarios.

## 2.2 DIFFUSION MODELS

Diffusion models (Sohl-Dickstein et al., 2015; Ho et al., 2020) have shown remarkable versatility in tasks such as class-conditioned generation, text-to-image synthesis, and image editing (Dhariwal & Nichol, 2021; Ho & Salimans, 2021; Meng et al., 2021; Ho et al., 2022; Saharia et al., 2022b; Ramesh et al., 2022; Nichol et al., 2022; Saharia et al., 2022a). Latent Diffusion Models (LDMs) (Rombach et al., 2022) address the computational complexity of diffusion models by operating in a lower-dimensional latent space through the use of a pre-trained autoencoder. This approach significantly reduces computational load while maintaining generation quality. LDMs also introduce a flexible conditioning mechanism through cross-attention layers, enabling integration of various conditioning information, such as text embeddings. Recent advances such as T2I adapters (Mou et al., 2024; Biner et al., 2024) further enhance the adaptability of LDMs. By introducing additional adapter cross-attention layers, these methods allow fine-tuning for new conditional tasks while keeping the base LDM frozen, reducing computational costs and data requirements for model adaptation.

## 2.3 OBJECT-CENTRIC LEARNING METHODS WITH DIFFUSION MODELS

Recently, several works have investigated the use of diffusion models in object-centric learning, such as Latent Slot Diffusion (LSD) (Jiang et al., 2023), SlotDiffusion (Wu et al., 2023b), (Jung et al., 2024) and GLASS (Singh et al., 2024). All these methods employ LDMs (Rombach et al., 2022) as slot decoders. Their approaches are similar in that they all use slot attention (Locatello et al., 2020) to extract slot representations from input image and then condition the diffusion model with these slots through cross-attention modules. These methods primarily differ in whether they use pretrained diffusion models, fine-tune them, or train them from scratch. LSD and GLASS use pretrained diffusion models to be able to deal with complex real-world images, assuming an inherent alignment between slot representations and the text-trained cross-attention within the diffusion model, which hinders their performance, especially on image generation tasks. SlotDiffusion, on the other hand, opts to retrain the diffusion model on the target dataset, thereby avoiding biases due to text-trained conditioning, though at the cost of increased computational complexity and, more importantly, at the loss of the generative power of pretrained diffusion models.

GLASS, which is actually a concurrent work (Singh et al., 2024), attempts to mitigate the issues due to text-trained conditioning in pretrained diffusion models by utilizing cross-attention masks as pseudo-ground truth to guide slot attention, in a manner similar to our approach. However, it necessitates an external image captioner or class labels to do this and requires two forward passes: one for generating the training images and pseudo masks, and another for slot attention and image reconstruction. Despite this, GLASS still struggles, particularly in differentiating object instances of the same class, and does not present any compositional image generation results. In contrast, SlotAdapt incorporates additional cross-attention layers into the pretrained diffusion model as adapters, enabling the slots to focus primarily on object semantics, rather than being constrained within a text-centric embedding space.

Jung et al. (2024) have recently introduced a novel framework for compositional learning, which incorporates an additional compositional path into their architecture alongside the slot-conditioned diffusion model. Their approach processes two images simultaneously, composing objects from these images into a single output to apply a diffusion-based generative prior loss. This facilitates the learning of compositional information; however, the applicability of the resulting model on complex real-world images is limited; more specifically, they present results on BDD100K (Yu et al., 2020), an autonomous driving dataset with limited context and relatively small number of object classes.

## 3 SLOT-BASED OBJECT-CENTRIC LEARNING

### 3.1 BACKGROUND

**Slot Attention:** Slot Attention (Locatello et al., 2020) provides a robust framework for segmenting input data into discrete, interpretable components. It operates by dynamically allocating a set of learnable vectors, termed "slots", to represent distinct entities within the input data. These slots

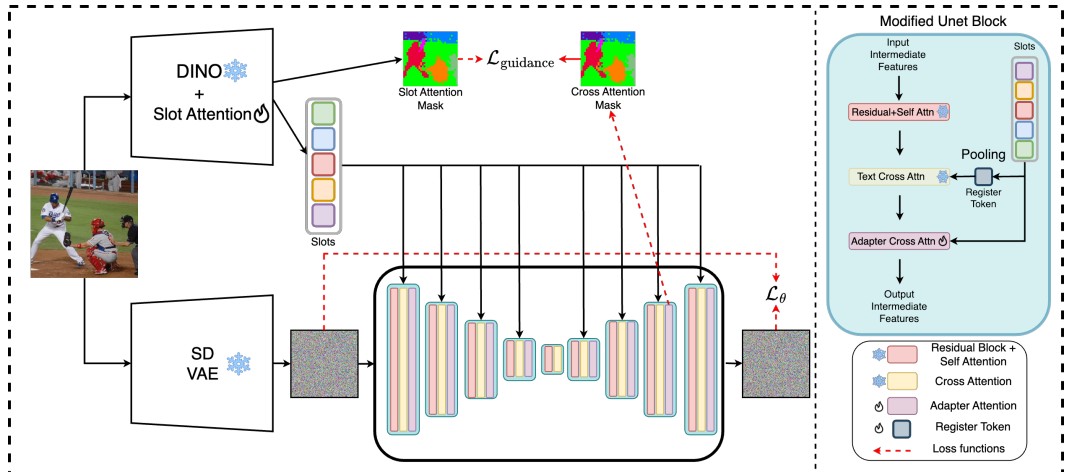

Figure 1: **SlotAdapt Architecture** We extract object-centric information from the input image using a visual backbone, which combines DINO and slot attention. Stable Diffusion VAE is used to encode the image into latent space and then noise is added to the latent. Diffusion process is conditioned on the generated slots as well as the register token which is generated by (mean) pooling the slots. We use the original cross attention layers of diffusion model to condition on the register token, and additional adapter attentions to condition on the learned slots. The overall objective is to predict the noise added to the image. Additionally, we introduce a guidance loss between the slot attention masks and adapter cross attention masks, which encourages the similarity between them. The guidance is only applied in the third upsampling block, while slot conditioning is applied throughout all downsampling and upsampling blocks.

are often initialized randomly and then iteratively updated via an attention mechanism, enabling the model to bind each slot to different parts of the input corresponding to individual objects or object-like features. The iterative update rule for each slot is given by

$$\mathbf{U}^{(m)} = \text{Attention}\left(q(\mathbf{S}^{(m)}), k(\mathbf{f}), v(\mathbf{f})\right) \tag{1}$$

$$\mathbf{S}^{(m+1)} = \text{GRU}(\mathbf{S}^{(m)}, \mathbf{U}^{(m)})$$

where $q$, $k$, $v$ are learnable linear functions, respectively corresponding to the query, key and value in attention computation; $\mathbf{f}$ denotes image features; $\mathbf{S}$ is the set of slots; $\mathbf{U}$ represents the update generated by the attention operation; and $m$ is iteration number for the GRU (Gated Recurrent Unit). The initial slots, $\mathbf{S}^{(0)}$, can be initialized from a Gaussian distribution or using initial object locations (Kipf et al., 2021; Elsayed et al., 2022). Attention is computed over the slot dimension, causing slots to compete for pixels. As a result, each slot attempts to bind to a distinct object in each iteration. The effectiveness of Slot Attention lies in its capacity to disentangle and encode complex scenes into structured representations without supervision. This is achieved through the ability of the mechanism to assign representational capacity where needed.

**Diffusion Models:** Diffusion models (Sohl-Dickstein et al., 2015; Ho et al., 2020) are a powerful class of generative models that simulate the gradual addition of noise to data and then learn to reverse this process. Once trained, they can generate high-quality samples from pure Gaussian noise, and have shown remarkable success in various domains, particularly in computer vision for challenging tasks such as image generation and editing.

The generative process involves a series of reverse diffusion steps from a noise distribution $p(\mathbf{z})$ to the data distribution $p(\mathbf{x})$, defined by conditional probabilities $p(\mathbf{x}_{t-1}|\mathbf{x}_t)$. The training objective minimizes the difference between the actual and generated distributions, typically using a loss function $\mathcal{L}$ that penalizes the expected error of noise prediction at each time step $t$:

$$\mathcal{L}(\boldsymbol{\theta}) = \mathbb{E}_{\mathbf{x}\sim p(\mathbf{x}), \boldsymbol{\epsilon}_t \sim \mathcal{N}(0,1), t\sim\mathcal{U}(1,T)}\left[\|\boldsymbol{\epsilon}_t - \boldsymbol{\epsilon}_{\boldsymbol{\theta}}(\mathbf{x}_t, t, y)\|_2^2\right], \tag{2}$$

where $\boldsymbol{\epsilon}_{\boldsymbol{\theta}}$ is the noise prediction of the neural network with parameters $\boldsymbol{\theta}$, and $y$ is a conditioning signal for the model such as class or text. When dealing with images, the common practice is to perform training and sampling in a low-dimensional latent space, which is achieved by first transforming the input via a pretrained variational encoder and then to use the standard U-Net architecture as the denoising neural network $\boldsymbol{\epsilon}_{\theta}$ (Rombach et al., 2022).

Previous OCL methods which employ diffusion models as slot decoders replace the condition $y$ in Eq. 2 with slot representations generated by Slot Attention (Locatello et al., 2020) and use reconstruction loss as a learning signal.

## 3.2 SLOTADAPT

**Object-centric Visual Encoding:** We start with an input image $\mathbf{x} \in \mathcal{R}^{H \times W \times 3}$ and transform it through a visual backbone combined with slot attention in a manner similar to recent works (Jiang et al., 2023; Wu et al., 2023b). The visual backbone serves as a feature extractor, reducing the image size to create a set of visual feature vectors $\mathbf{f} \in \mathcal{R}^{h \cdot w \times d}$ by flattening. This reduction makes computations more efficient and condenses key features of the image into a more compact representation. The slot attention mechanism works on these vectors, using a competitive process to generate $N$ slots, represented as $\mathbf{S} \in \mathcal{R}^{N \times d}$. Each slot contains information about a separate object or entity in the scene. The resulting slot representations are then used to condition the diffusion-based decoder as explained in the following section (see also Fig. 1).

**Slot Conditional Decoding:** We employ a pretrained Stable Diffusion UNet model as a decoder conditioned on the extracted slot representations. This UNet is primarily composed of residual, self-attention, and cross-attention layers stacked together. While self-attention layers capture spatial information, cross-attention layers model semantic relationships between text embeddings and intermediate features. Inspired by recent works (Mou et al., 2024; Ye et al., 2023), we extend this architecture by introducing an adapter layer after each existing cross-attention layer in all downsampling and upsampling blocks of the UNet. These adapter layers are dedicated to conditioning the model on the extracted slots and are essentially cross-attention layers which function similarly to their text-based counterparts in the UNet, with a few minor differences (see Appendix for implementation details).

Our adapter-based conditioning strategy is substantially different from the conventional use of text-trained cross-attention for slot conditioning. Our rationale is that by including these additional cross-attention layers, we enable the slots to focus primarily on object semantics, rather than being constrained within a text-centric embedding space. This is particularly crucial, given that the cross-attention layers in pretrained diffusion modules are typically optimized for text embeddings and hence expect textual input.

We also introduce an extra token that leverages the unused text-conditioning modules of the pretrained UNet to better capture context. This extra token is computed by (mean) pooling either the generated slots or the image features from the visual backbone, and then fed as input into the text cross-attention layers of the UNet model. The main motivation here is to create a 'register' token, similar to the idea presented by Darcet et al. (2024). This register acts as a storage for overall scene information within the diffusion model. By giving this task to a dedicated token, we allow the individual slots to concentrate more on specific objects instead of diluting their focus with background or contextual details.

We freeze the pretrained diffusion model and train only the adapter layers and the slot attention component of our architecture. The training process minimizes a reconstruction objective, formulated as a noise prediction problem. Following (Rombach et al., 2022), at each training iteration (assuming a mini-batch of size 1), time step $t$, latent image $\mathbf{x}$ and noise $\epsilon_t$ are first sampled from their respective distributions. The noisy image $\mathbf{x}_t$ is then calculated via forward diffusion and fed into the denoising UNet $\epsilon_{\boldsymbol{\theta}}$. The UNet predicts the noise $\epsilon_t$, conditioned on the extracted slots $\mathbf{S}$ and the register token $\mathbf{r}$, and is updated based on the gradient of the following loss function (see also Eq. 2):

$$\mathcal{L}_t(\boldsymbol{\theta}) = \|\epsilon_t - \epsilon_{\boldsymbol{\theta}}(\mathbf{x}_t, t, \mathbf{S}, \mathbf{r})\|_2^2 \tag{3}$$

**Attention Guidance:** The attention mask generated by slot attention serves as an affinity measure between image features and slot vectors, effectively segmenting objects in the image under the assumption that the slots capture object representations. We enhance this framework by leveraging the cross-attention mask extracted from the adapter layers in the diffusion model as a self-supervisory signal to guide slot attention, and/or vice versa. These dual attention masks, one from slot attention and the other from the diffusion model, encode similar semantics regarding the relationship between slot representations and image features. Normally, there are as many adapter cross-attention masks as there are UNet blocks, but we focus only on the one from the third upsampling block (the second-to-last) since this layer is very close to the output and empirically provides an attention mask most

aligned with the objects in the image. We denote the slot attention mask by $\mathbf{A}_{\text{SA}}$ and the diffusion attention mask with $\mathbf{A}_{\text{DM}}$. In both cases, the attention masks are computed through the dot product between queries and keys, and normalized over the query dimension. In slot attention, slots act as queries and image features as keys, whereas in the attention mechanism of the diffusion model, this relationship is inverted. Formally we can write

$$\mathbf{A}_{\text{SA}} = \text{Softmax}\left(\frac{k_{\text{SA}}(\mathbf{f}_{\text{SA}})q_{\text{SA}}(\mathbf{S})^\top}{\sqrt{D}}\right) \qquad \mathbf{A}_{\text{DM}} = \text{Softmax}\left(\frac{k_{\text{DM}}(\mathbf{S})q_{\text{DM}}(\mathbf{f}_{\text{DM}})^\top}{\sqrt{D}}\right) \qquad (4)$$

where $q_{\text{SA}}$, $k_{\text{SA}}$, $q_{\text{DM}}$, and $k_{\text{DM}}$ are learnable linear functions, and $\mathbf{f}_{\text{SA}}$ and $\mathbf{f}_{\text{DM}}$ represent image features. The cross-attention layer in the adapter employs a multi-head structure, resulting in multiple attention masks. We average these masks over the head dimension and use the result as the diffusion attention mask $\mathbf{A}_{\text{DM}}$.

The slot attention mask, $\mathbf{A}_{\text{SA}} \in \mathcal{R}^{(h \cdot w) \times N}$, encodes how each pixel in the image features relates to the slot representations. Ideally, this attention mask should converge to an instance segmentation mask, with each slot representing a distinct object. Conversely, the diffusion attention mask, $\mathbf{A}_{\text{DM}} \in \mathcal{R}^{N \times (h \cdot w)}$, shows the inverse relationship. In the optimal scenario, we expect this attention map to converge to the transpose of the instance segmentation mask, enabling the diffusion model to generate the input image accurately. We formulate a guidance loss to enforce the alignment of these dual attention masks by

$$\mathcal{L}_{\text{guidance}} = \text{BCE}(\mathbf{A}_{\text{SA}}, \mathbf{A}_{\text{DM}}^\top), \qquad (5)$$

where BCE is the binary cross-entropy loss. The overall training objective combines this guidance loss with the primary loss function, weighted by a hyperparameter $\lambda$:

$$\mathcal{L} = \mathcal{L}_{\boldsymbol{\theta}} + \lambda \mathcal{L}_{\text{guidance}} \qquad (6)$$

There are different design choices for implementing the guidance loss: 1) guiding only the slot attention mask $\mathbf{A}_{\text{SA}}$ with $\mathbf{A}_{\text{DM}}$ (stopping gradient for diffusion model), 2) guiding only the diffusion attention mask $\mathbf{A}_{\text{DM}}$ with $\mathbf{A}_{\text{SA}}$ (stopping gradient for slot attention), and 3) joint guidance, that is, guiding both $\mathbf{A}_{\text{SA}}$ and $\mathbf{A}_{\text{DM}}$ simultaneously (no gradient stopping). So in the first two options, one attention mask serves as a pseudo ground-truth for the other, whereas in the last option, the two masks are jointly enforced for alignment.

Another alternative we have considered is guiding the attention masks through multiplication. In this scenario, there is no explicit auxiliary loss: each of the cross-attention masks from multiple attention heads in the adapter layer is simply multiplied by $\mathbf{A}_{\text{SA}}$ using the Hadamard product. Since each cross-attention mask ideally represents a part of the object bound by the corresponding slot, the multiplication confines the adapter attention to the region defined by the slot attention mask.

All the guidance alternatives described above aim, in one way or another, to enhance the alignment of the attention matrices which represent the same semantics in the architecture, thereby improving their alignment with the objects in the input image. We compare these different alternatives for attention mask guidance in the experiments section.

## 4 Experiments

**Datasets:** Our evaluation framework covers both synthetic and real-world datasets, aligning with recent works in the field (Jiang et al., 2023; Wu et al., 2023b). We assess our method SlotAdapt on the synthetic MOVi-E dataset (Greff et al., 2022) and two widely-recognized real-world datasets: VOC (Everingham et al., 2010) and COCO (Lin et al., 2014). While our primary focus is on leveraging pretrained diffusion models for real-world scenarios, we include MOVi-E in our evaluation due to its complexity, featuring scenes with up to 23 objects. This dataset serves as a challenging benchmark for object-centric learning in controlled environments. Both real-world datasets, VOC and COCO, have emerged as popular benchmarks for object discovery tasks. They present significant challenges due to their multi-object nature and the large number of foreground classes they contain—20 and 80, respectively. These datasets have been instrumental in recent evaluations of various object-centric learning methods on real-world images (Jiang et al., 2023; Wu et al., 2023b; Seitzer et al., 2023).

**Implementation Details:** Following the previous works (Jiang et al., 2023; Wu et al., 2023b), we use a convolutional backbone for MOVi-E and DINOv2 (Oquab et al., 2023) with ViT-B and a patch

size of 14 as the encoder model for VOC and COCO. To serve as our decoder, we incorporate a pretrained Stable Diffusion (SD) model, v1.5 for MOVi-E; and COCO and v2.1 for VOC (Rombach et al., 2022), augmented with an additional cross-attention layer as adapter. For MOVi-E, we jointly optimize the image backbone, slot attention mechanism, and adapter layers. For VOC and COCO, we focus our training exclusively on the slot attention and adapter layers. We maintain consistency with previous work (Jiang et al., 2023; Wu et al., 2023b) in terms of dataset selection and preparation. Our models are trained for approximately 150K to 250K iterations. While this training duration is shorter compared to some previous works, we demonstrate that our approach achieves competitive performance.

Table 1: **Architectural ablations on MOVi-E.** We examine the effects of architectural choices on segmentation and representation performance. We present block combinations on the left and register token choice on the right. Up, Down and Mid refer to all upsampling blocks, all downsampling blocks and middle block in the diffusion model.

| | Conditioning Blocks | | | | | Register Token | | |
|---|---|---|---|---|---|---|---|---|
| | Up+Down+Mid | Only Up | Only Down | Up+Mid | Up+Down | No Token | Slot Pooling | Feature Pooling |
| Segmentation (%) | | | | | | | | |
| FG-ARI (↑) | 56.89 | 57.38 | **57.93** | 57.39 | 56.45 | 54.38 | 56.27 | **57.18** |
| mBO (↑) | 39.59 | 43.05 | 40.20 | 39.96 | **43.38** | 40.07 | 41.65 | **43.98** |
| mIoU (↑) | 37.75 | 41.53 | 38.77 | 38.83 | **41.86** | 40.07 | **40.10** | 39.83 |
| Representation | | | | | | | | |
| Category (↑) | 43.92 | 43.82 | **45.88** | 41.54 | 43.91 | 42.42 | **43.54** | 42.63 |
| Position (↓) | 1.92 | 1.82 | **1.61** | 1.92 | 1.72 | 1.89 | **1.75** | 1.78 |
| 3D B-Box (↓) | 3.94 | 3.78 | **3.48** | 3.95 | 3.75 | 3.83 | **3.77** | 3.78 |

**Baselines:** We compare SlotAdapt with unsupervised state-of-the-art object-centric methods. On the MOVi-E dataset, we compare with SLATE (Singh et al., 2021), SLATE$^+$, where SLATE's low capacity dVAE is replaced by a pre-trained VQGAN model (Esser et al., 2021) and Latent Slot Diffusion (LSD) (Jiang et al., 2023). On real-world datasets, we compare with Slot attention (SA) (Locatello et al., 2020), SLATE (Singh et al., 2021), DINOSAUR (Seitzer et al., 2023), LSD (Jiang et al., 2023) and SlotDiffusion (Wu et al., 2023b). In all our experiments on real-world datasets, we use a frozen DINOv2 (Caron et al., 2021; Oquab et al., 2023) as the visual encoder for all models.

**Metrics:** Following previous work (Locatello et al., 2020; Jiang et al., 2023), we employ a set of standard metrics to assess the performance of our method on unsupervised object segmentation. These include the foreground adjusted rand index (FG-ARI), mean intersection over union (mIoU), and mean best overlap (mBO). Our evaluation is performed on the slot attention masks, $\mathbf{A}_{\text{slot}}$, computed as in Equation 4. We use two different versions of the mIoU and mBO metrics: one computed over instance-level masks, and the other over class-level masks. Note that instance-level metrics account for whether objects of the same class in an image are differentiated as separate instances in the resulting segmentation; hence, they are more informative on the representational and generative capabilities of an object-centric learning method. On the other hand, the FG-ARI, a metric designed primarily for object discovery task, does not account for object masks larger than the ground-truth, which may be problematic, especially when assessing the generative capability.

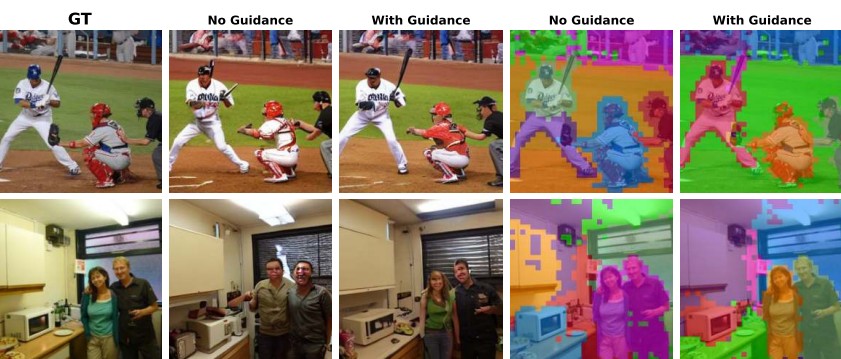

Figure 2: **Qualitative comparison: with vs. without guidance.** We visualize generated images and predicted segments on COCO dataset.

Table 2: **Evaluation of guidance strategies.** We present the segmentation performance on COCO for different guidance strategies. Joint guidance gives the best scores and significantly improves over no guidance option.

| | FG-ARI | mBO$^i$ | mBO$^c$ | mIoU$^i$ | mIoU$^c$ |
|---|---|---|---|---|---|
| No guidance | 42.3 | 31.5 | 34.8 | 31.7 | 38.5 |
| Slot Guidance | 41.2 | 33.4 | 36.9 | 33.1 | 37.9 |
| DM Guidance | 42.0 | 31.2 | 34.6 | 32.0 | 38.4 |
| Joint Guidance | 41.4 | **35.1** | **39.2** | **36.1** | **41.4** |
| Multiplication Guidance | **43.3** | 31.9 | 35.3 | 31.7 | 36.4 |

## 4.1 ABLATION STUDIES

To assess the impact of our contributions, we conduct a series of experiments. All the experiments in this ablation study are conducted with register token (slot pooling), joint guidance of attention masks and conditioning through all downsampling and upsampling blocks, unless stated otherwise.

We first investigate the optimal block combination in the UNet architecture for conditioning the diffusion model on slots, where the options we consider are all downsampling blocks, all upsampling blocks, mid-block and their certain combinations. The results obtained on MOVi-E dataset are presented in the left part of Table 1. We observe that conditioning on either the upsampling blocks alone or both the downsampling and upsampling blocks yield superior performance, likely due to their proximity to the input and output in terms of structure and resolution.

Next, we evaluate the effect of incorporating a register token (Darcet et al., 2024) into the textual cross-attention layer in the diffusion model. The results obtained on MOVi-E are given in the right part of Table 1), where the options are no register token, slot pooling and feature pooling. We observe that inclusion of register token, with slot or feature pooling, yields consistent improvements in all metrics, including segmentation accuracy and performance in downstream tasks.

Lastly, we examine the effectiveness of our guidance strategies on COCO dataset in Table 2. We observe that joint guidance yields the best performance on all metrics except the FG-ARI score, where the improvements are substantial compared to the no-guidance case. For FG-ARI metric, multiplication guidance gives the best performance, which is a sufficient metric for object discovery but not necessarily for generative tasks. Moreover, in Fig. 2, we visually demonstrate the impact of the joint guidance strategy on the generated segmentation masks. We observe that the inclusion of guidance yields a significant improvement in segmentation quality, mitigating the part-whole hierarchy problem and producing segmentation masks better aligned with the objects in the scene, rather than with partial or fragmented objects. In turn, the generated (reconstructed) images are more faithful to the original input images, as also observed in Fig. 2.

## 4.2 EVALUATION RESULTS

All the evaluation experiments for SlotAdapt are conducted with register token, joint guidance and conditioning on all downsampling and upsampling blocks, unless stated otherwise.

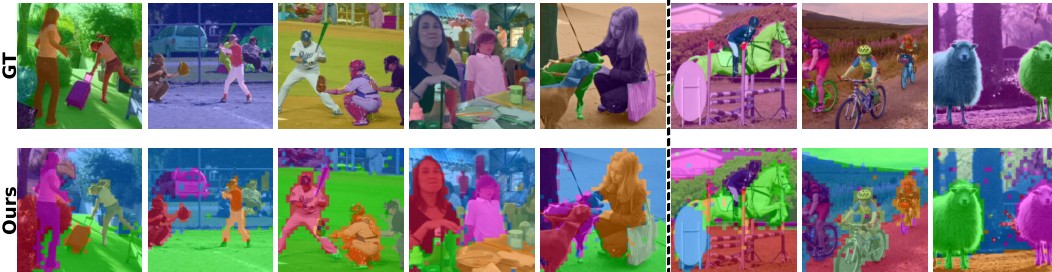

Figure 3: **Unsupervised Object Segmentation.** We show visualizations of segments on COCO (left) and VOC (right). SlotAdapt accurately binds distinct instances belonging to the same class.

**Synthetic Dataset:** We evaluate the object discovery and segmentation performance of our method on MOVi-E in comparison to the baseline methods in Table 5, which is presented in Appendix, based on the FG-ARI, instance-level mBO and instance-level mIoU metrics. We observe significant

Table 3: **Model Evaluation** Comparison of FID and KID scores for reconstruction (left) and compositional generation (right) across different methods.

| Method | FID | KID×1000 | Method | FID | KID×1000 |
|---|---|---|---|---|---|
| LSD | 35.537 | 19.086 | LSD | 167.232 | 103.482 |
| SlotDiffusion | 19.448 | 5.852 | SlotDiffusion | 64.213 | 57.309 |
| Ours | **10.857** | **0.388** | Ours | **40.568** | **34.381** |

improvements in almost all metrics, particularly with an approximate 10% enhancement in object discovery and segmentation accuracy.

**Real-World Dataset:** We evaluate the object discovery and segmentation performance of our method on the COCO and VOC datasets in Table 4, using the FG-ARI, instance-level mBO and class-level mBO metrics, in comparison to baseline methods including StableLSD (Jiang et al., 2023), SlotDiffusion (Wu et al., 2023b), and DINOSAUR (Seitzer et al., 2023).

We observe that SlotAdapt, when used with guidance, outperforms all the baselines for almost all metrics, including the highly competitive DINOSAUR method. The only exception is the class-level mBO score on VOC, where SlotAdapt performs worse than SlotDiffusion. Notably, the improvement in mBO scores on COCO is particularly significant, about 10% better than the next best baseline. The impact of guidance is also substantial on COCO, which is a more challenging and much larger dataset with complex multi-object scenes and varying object sizes, when compared to VOC.

In Fig. 4, we visually compare the segmentation results of LSD, SlotDiffusion and SlotAdapt on COCO. We observe that SlotAdapt successfully differentiates individual object instances as evidenced by its superior instance-level mBO score, whereas LSD and Slot Diffusion struggle with this challenge. Moreover, SlotAdapt produces more complete segmentations of objects without dividing them into parts, which is reflected by its higher FG-ARI score. These results highlight the robustness and versatility of SlotAdapt in handling the complexities of real-world data.

**Generation and Compositional Editing:** We first demonstrate the ability of our model to generate realistic images on COCO in Fig. 5. We see that, when conditioned on slots, our model can reconstruct the input image with high quality, realism and notable fidelity. Regarding compositional generation and editing capabilities, Figure 6 shows a series of image edits by modifying input slots, including object replacement, removal and addition. We observe that the editing operations are highly successful and seamless with only slight yet realistic changes to the image background, while all maintaining high quality. To assess our model's capabilities, we conducted extensive experiments evaluating both its reconstruction fidelity and compositional generation performance. The comprehensive results are presented in Table 3. For additional qualitative and quantitative results, we refer readers to Section A.3 in the appendix.

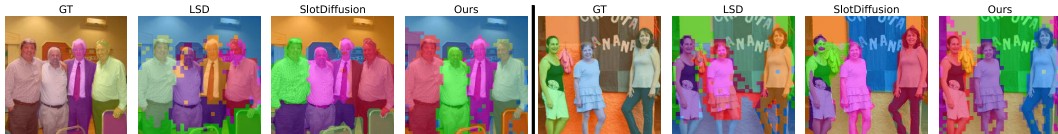

Figure 4: **Qualitative comparisons with other methods on COCO.** We visualize predicted segments of SlotAdapt in comparison to LSD and SlotDiffusion. SlotAdapt can more effectively differentiate between object instances of the same class compared to other methods.

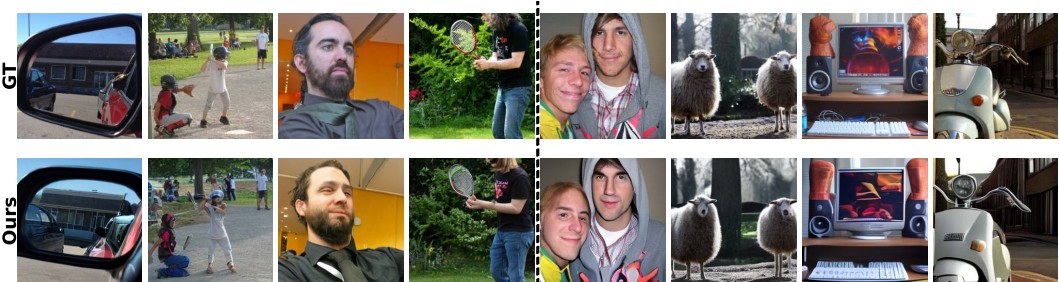

Figure 5: **Generation Results.** We show sample images reconstructed by SlotAdapt on COCO (left) and VOC (right). SlotAdapt generates reconstructions highly faithful to the original input images.

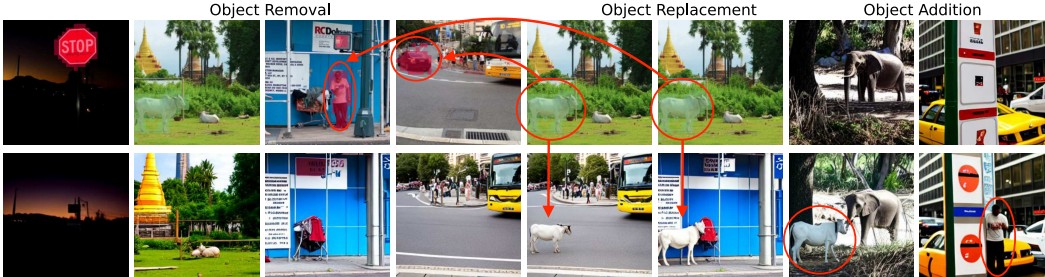

Figure 6: **Compositional Editing.** We demonstrate object removal, replacement and addition edits on COCO images by using slots. Removing highlighted slots (top row) yields realistic and successful generations (first 4 examples). Replacing highlighted objects in the 3rd and 4th images with the cow object from the 5th and 6th images results in highly accurate edits, yet with small changes in the original images. Finally, adding the cow (5th image) and the person (3rd image) slots to the last two images, respectively, generates meaningful examples of complex scenes.

Table 4: **Unsupervised object segmentation on real-world datasets.** We compare SlotAdapt with state-of-the-art methods on VOC (left) and COCO (right). We present two versions of our method, with and without guidance loss.

| VOC | FG-ARI | mBO$^i$ | mBO$^c$ | COCO | FG-ARI | mBO$^i$ | mBO$^c$ |
|---|---|---|---|---|---|---|---|
| SA + DINO ViT | 12.3 | 24.6 | 24.9 | SA + DINO ViT | 21.4 | 17.2 | 19.2 |
| SLATE + DINO ViT | 15.6 | 35.9 | 41.5 | SLATE + DINO ViT | 32.5 | 29.1 | 33.6 |
| DINOSAUR | 23.2 | 43.6 | 50.8 | DINOSAUR | 34.3 | 32.3 | 38.8 |
| LSD | 18.7 | 40.5 | 43.5 | LSD | 33.8 | 27.0 | 30.5 |
| SlotDiffusion | 17.8 | 50.4 | **55.3** | SlotDiffusion | 37.2 | 31.0 | 35.0 |
| Ours | 28.8 | **51.6** | 52.0 | Ours | **42.3** | 31.5 | 34.8 |
| Ours + Guidance | **29.6** | 51.5 | 51.9 | Ours + Guidance | 41.4 | **35.1** | **39.2** |

## 5 CONCLUSION

We have targeted the object-centric learning problem, particularly on complex real-world images. To this end, we have presented a method that combines slot attention with pretrained diffusion models. Our architecture, SlotAdapt, has three main novelties; adapters in the diffusion model for slot conditioning, the use of a register token to represent background in images, and attention guidance to align slot attention with cross attention from the diffusion model. We have conducted extensive experiments, particularly on COCO dataset, to validate our approach. Our experiments show that leveraging the generative capabilities of pretrained diffusion models is crucial for tackling the challenging tasks such as object discovery, unsupervised segmentation and compositional generation and editing on complex real-world images. Our method takes a step forward in achieving this goal, outperforming the state of the art methods on the challenging COCO dataset without relying on any external supervision in object discovery, segmentation and, particularly compositional generation and editing. We are the first to present successful compositional editing experiments on the COCO dataset.

A limitation of our method from the perspective of compositional generation is that the edited or reconstructed images, when conditioned on slots, may exhibit slight changes with respect to the source image, though the generations are mostly highly realistic and of very good quality. One remedy for this can be the incorporation of additional training objectives to enforce fidelity to the input. Another related issue is how to make use of this framework for image editing in practice, which needs further adjustments to the architecture possibly to accept user prompts and associate them with slot representations. Lastly, our method still has issues for under- and over-segmentation, which may potentially be mitigated through slot merging and splitting. Additionally, our approach relies on pre-trained diffusion models that are predominantly trained on real-world data, which may limit their adaptability to synthetic domains. While the model performs well on natural images, its effectiveness may be reduced when handling synthetic datasets such as CLEVR and CLEVRTex Johnson et al. (2017).

**Acknowledgements:** We thank Aykut Erdem and Erkut Erdem for their insightful discussions.

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
