# A  APPENDIX

In this appendix, we provide more details and results about our work. It is organized into three parts:

- Training and Architectural Details (section A.1): We explain how we trained our model SlotAdapt and describe its structure in depth.
- Additional Experiments and Evaluations (section A.2): We share additional experimental and evaluation results
- More Examples (section A.3): We show additional visual examples of SlotAdapt's performance in different settings.
- Comparison with GLASS and SPOT (section A.4): We compare SlotAdapt with GLASS and SPOT, similar studies, to show how our work fits into current research.

## A.1  TRAINING AND ARCHITECTURAL DETAILS

This section provides comprehensive information on the architectural components and training procedures of our model.

The importance of object-centric representations has been recognized across different domains, emerging as a future direction in works on temporal prediction Akan et al. (2021) and autonomous driving Akan & Güney (2022). Building on this insight, recent approaches have successfully applied diffusion models to achieve object-centric learning Jiang et al. (2023); Wu et al. (2023b) and and our approach enhances this paradigm through more effective slot-based representations.

### A.1.1  ARCHITECTURAL DETAILS

Figure 1 in the main text illustrates our training pipeline. Below, we detail the key components:

**Diffusion Model:**  We initialize a UNet denoiser and VAE from pretrained diffusion models (Rombach et al., 2022). Specifically:

- MOVi-E (Greff et al., 2022) and COCO Lin et al. (2014) datasets: Stable Diffusion v1.5
- VOC (Everingham et al., 2010): Stable Diffusion v2.1

After initialization, we inject adapter layers following each downsampling and upsampling block in the UNet. Each adapter layer comprises a cross-attention mechanism and a feedforward network, both preceded by layer normalization.

**VAE:**  We employ pretrained VAEs, maintaining consistency with the diffusion model versions: Stable Diffusion v1.5 VAE for MOVi-E and COCO; Stable Diffusion v2.1 VAE for VOC

**Visual Encoder Backbone:**

- MOVi-E: We utilize a custom CNN backbone encoder. It consists of 4 downsampling blocks, 1 middle block, and 4 upsampling blocks, with channel multipliers [1,1,2,4] and a base channel count of 128. Each block contains 2 residual layers and the overall network's output channels 128 channels.
- COCO and VOC: We employ a frozen DINOv2 (Oquab et al., 2023) model with a ViT-B backbone (patch size 14).

The extracted feature maps maintain a consistent resolution of 32×32 across all datasets.

**Slot Attention:**  Our implementation varies by dataset:

- MOVi-E: We adopt the LSD (Jiang et al., 2023) configuration (slot size: 192, iterations: 3, slots: 24). We append four linear projectors to align slot dimensions with adapter attention layers.

Table 5: **Comparative evaluation on MOVi-E:** (Left) Segmentation results, (Right) Representation assessment: We evaluate slots through predictive probing. Spatial attributes (position, 3D bounding box) are assessed via MSE (mean squared error), while categorical predictions are assessed by classification accuracy.

| Segmentation | SLATE | SLATE$^+$ | LSD | Ours | Representation | SLATE | SLATE$^+$ | LSD | Ours |
|---|---|---|---|---|---|---|---|---|---|
| mBO ($\uparrow$) | 30.17 | 22.17 | 38.96 | **43.38** | Position ($\downarrow$) | 2.09 | 2.15 | 1.85 | **1.77** |
| mIoU ($\uparrow$) | 28.59 | 20.63 | 37.64 | **41.85** | 3D B-Box ($\downarrow$) | 3.36 | 3.37 | **2.94** | 3.75 |
| FG-ARI ($\uparrow$) | 46.06 | 45.25 | 52.17 | **56.45** | Category ($\uparrow$) | 38.93 | 38.00 | 42.96 | **43.92** |

- COCO and VOC: We use a slot size of 768 and 7 slots, with similar output projector layers as in MOVi-E.

For all datasets, we use a linear layer to project either pooled visual backbone features or averaged slot vectors to a 768-dimensional space to match the text cross-attention dimensions in the diffusion model. We use slot averaging for MOVi-E and COCO datasets, and feature pooling for the VOC dataset, as they perform slightly better.

**Training Details:**

- Hardware: 2 NVIDIA A40 GPUs
- Batch sizes: 32 (MOVi-E), 30 (VOC), 32 (COCO)
- Training iterations: 200K (MOVi-E), 250K (COCO), 190K (VOC)
- Optimization: AdamW optimizer, constant learning rate, FP16 precision
- $\lambda$: In first 40K iterations, we use 0, from 40K-50K, it is increased from 0 to the 0.025, then, we use 0.025. Since the slots and adapter layers are initialized completely random, we opt to wait for 40K iterations so that the attention masks have meaningful connections.

**Experimental Details:** For our experiments, we first select the arhictectural details (conditioning place and register token type) on MOVi-E dataset. Then, we migrate that architecture to the real-world setup.

**Dataset Details:** For **MOVi-E**, we follow the train-test split in Singh et al. (2022) and use $256 \times 256$ resolution for both diffusion model and slot attention. For **COCO**, we follow Seitzer et al. (2023) to train on the same training set with DINOSAUR, LSD and Slot Diffusion, which consists of 118,287 images for training and 5000 for validation. For **VOC**, we follow Seitzer et al. (2023) to train on the "trainaug" set so that the training sets are the same as DINOSAUR, LSD and Slot Diffusion. The training set contains 10,582 images and validation set contains 1449 images. For COCO and VOC, we use random horizontal flip in training as data augmentation and we use $256 \times 256$ for diffusion model and $448 \times 448$ for slot attention.

## A.2 ADDITIONAL EXPERIMENTS

**Synthetic Dataset Results:** We evaluate the object discovery and segmentation performance of our method on MOVi-E in comparison to the baseline methods in Table 5 based on the FG-ARI, instance-level mBO and instance-level mIoU metrics. To measure the representational quality of the learned slots for downstream tasks, we implement a straightforward approach using a 2-layer MLP to predict discrete categories, object positions and 3D bounding boxes. We also display the resulting prediction accuracies in Table 5. We observe significant improvements in almost all metrics, particularly with an approximate 10% enhancement in object discovery and segmentation accuracy.

**Better Segmentation Model:** In this part, we replace Slot Attention with BOQ-SA, which is an improved version of slot attention where the slot initialization is also optimized, Jia et al. (2023). The results (in Table 6) show that using BOQ-SA instead of Slot Attention results in modest improvements across several metrics.

**Slot Number:** We investigate the impact of slot count by comparing our original configuration of 7 slots with an increased count of 80 slots, departing from the convention established in previous works

Table 6: **Impact of a Better Segmentation Method:** We evaluate the effect of a better segmentation model on performance by replacing Slot Attention with BOQ-SA, an improved version. Results are presented on the COCO dataset.

| Method | FG-ARI | mBO$^i$ | mIoU$^i$ | mBO$^c$ | mIoU$^c$ |
|---|---|---|---|---|---|
| Slot Attention | 42.3 | 31.5 | 31.7 | 34.8 | 38.5 |
| BOQ-SA | 42.2 | 31.2 | 32.551 | 35.266 | 37.82 |

Seitzer et al. (2023); Wu et al. (2023b); Jiang et al. (2023) where slot count typically matches the maximum number of objects per scene in a given dataset. Our experiments revealed that dramatically increasing the slot count to 80 led to degraded performance, presented in Table 7.

**Starting Guidance Loss from the start:** We also experiment with starting of guidance loss, where the original idea was starting the guidance loss after some iterations so that the slots and the adapters have a knowledge of the objects. Below, we present results where the guidance loss is started in the beginning without waiting slots and adapters learn meaningful features. The results show that starting guidance loss after some iterations leads to better performance, presented in Table 8.

**Additional Token for Slot Attention as a register token:** We investigate an alternative approach to capturing global scene information in our architecture. Instead of using the average of all slot tokens as a global register token, we modified the slot attention module to include an additional dedicated slot specifically designed to capture global scene context. This dedicated slot was then directly used in the cross-attention layer. Our experimental results revealed interesting trade-offs: while the dedicated global slot approach improved foreground segmentation quality, it showed slightly decreased performance in semantic segmentation and generation quality. We hypothesize that this performance difference stems from the competitive nature of attention in the slot attention module—the dedicated global slot must compete with object slots for attention weights, potentially limiting its capacity to capture comprehensive scene information compared to our original approach of averaging all slot representations. This suggests that using the collective information from all slots through averaging provides a more robust global representation than a single learned global token, presented in Table 9

**Classifier-free Guidance:** We conduct an ablation study for classifier-free guidance value, which simply increases the effect of the conditioning in diffusion models Ho & Salimans (2022). We find that CFG value of 1.3 results in the best results in terms both FID Heusel et al. (2017) and KID Bińkowski et al. (2018), presented in Table 10

Table 7: **Impact of Slot Count on Performance:** We analyze the effect of varying the number of slots on the COCO dataset.

| Method | FG-ARI | Instance | | Semantic | | FID | KID |
|---|---|---|---|---|---|---|---|
| | | mBO | mIoU | mBO | mIoU | | |
| 80 slots | 18.1 | 23.2 | 26.4 | 28.6 | 32.9 | 114.236 | 64.610 |
| 7 slots | 41.4 | 35.1 | 36.1 | 39.2 | 41.4 | 10.857 | 0.388 |

Table 8: **Impact of Guidance Loss Timing:** We compare the effect of applying the guidance loss from the start of training versus introducing it after the model has learned initial representations. Results are presented on the COCO dataset.

| Method | FG-ARI | Instance | | Semantic | |
|---|---|---|---|---|---|
| | | mBO | mIoU | mBO | mIoU |
| Start from 0 | 37.85 | 32.65 | 33.99 | 36.059 | 39.254 |
| Original (start after 40K) | 41.4 | 35.1 | 36.1 | 39.2 | 41.4 |

Table 9: **Impact of Global Information Capturing Strategies:** We compare slot averaging and the use of an additional slot token for capturing global information. Results are presented on the COCO dataset.

| Method | FG-ARI | Instance | | Semantic | | FID | KID |
|---|---|---|---|---|---|---|---|
| | | mBO | mIoU | mBO | mIoU | | |
| Additional Slot Token | 43.8 | 31.9 | 32.4 | 35.5 | 37.3 | 11.212 | 0.431 |
| Slot Average Token | 42.3 | 31.5 | 34.8 | 34.8 | 38.5 | 10.857 | 0.388 |

Table 10: **Impact of CFG Value on Generation Quality:** We evaluate the effect of different CFG values on generation quality using the COCO dataset.

| CFG Value | FID | KID$\times$1000 |
|---|---|---|
| 7.5 | 21.350 | 6.271 |
| 5.0 | 17.558 | 4.236 |
| 2.5 | 13.734 | 2.151 |
| 2.0 | 12.427 | 1.424 |
| 1.5 | 11.041 | 0.590 |
| 1.4 | 10.880 | 0.459 |
| 1.3 | **10.857** | **0.388** |
| 1.2 | 11.057 | 0.492 |

## A.3 ADDITIONAL QUALITATIVE EXAMPLES

In this section, we present additional qualitative examples to further illustrate the capabilities of our proposed SlotAdapt model. These results complement the main paper by providing a more comprehensive view of our model's performance across various tasks and datasets.

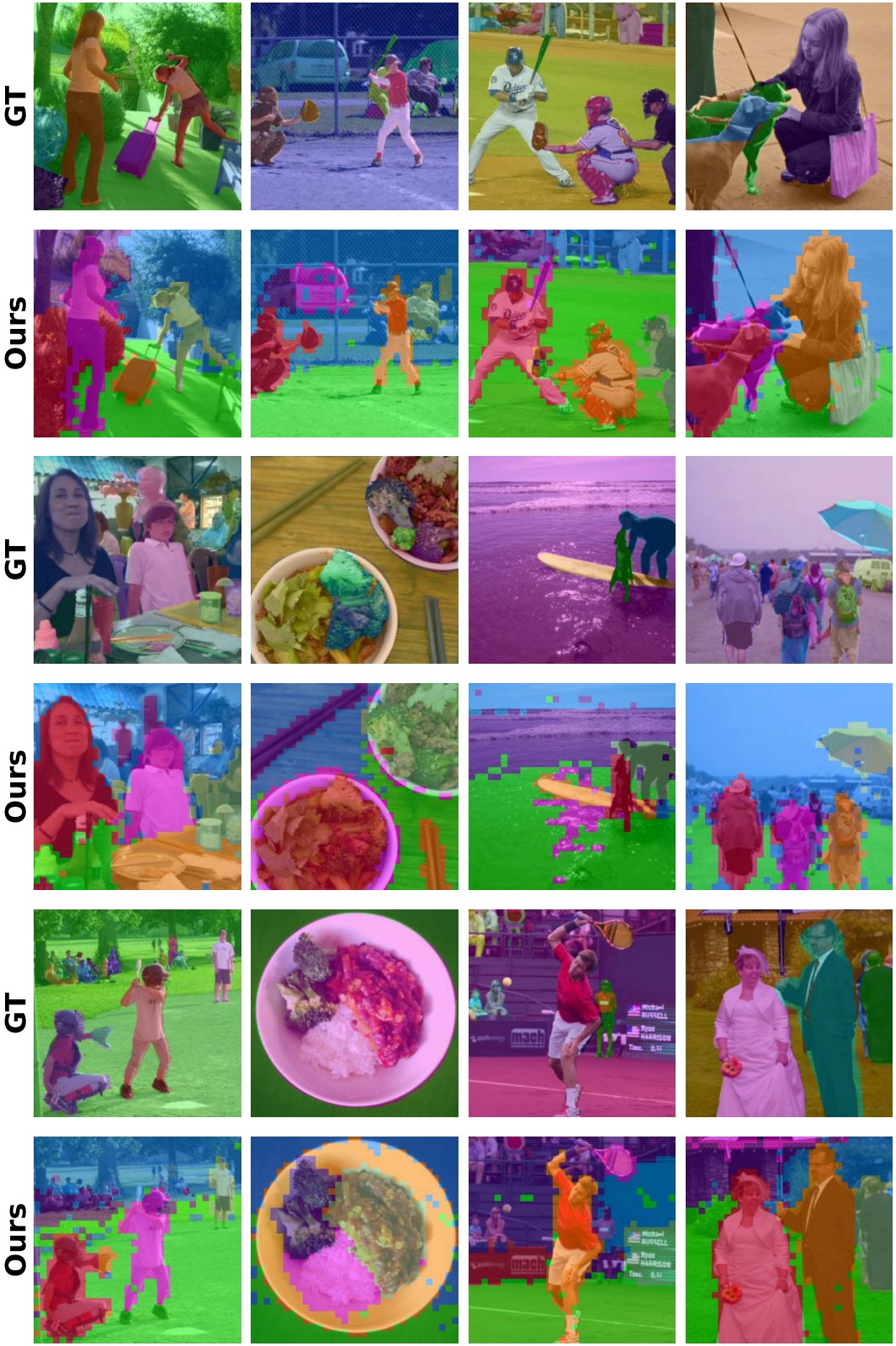

Figure 7: **Unsupervised Object Segmentation.** We show visualizations of predicted segments on COCO.

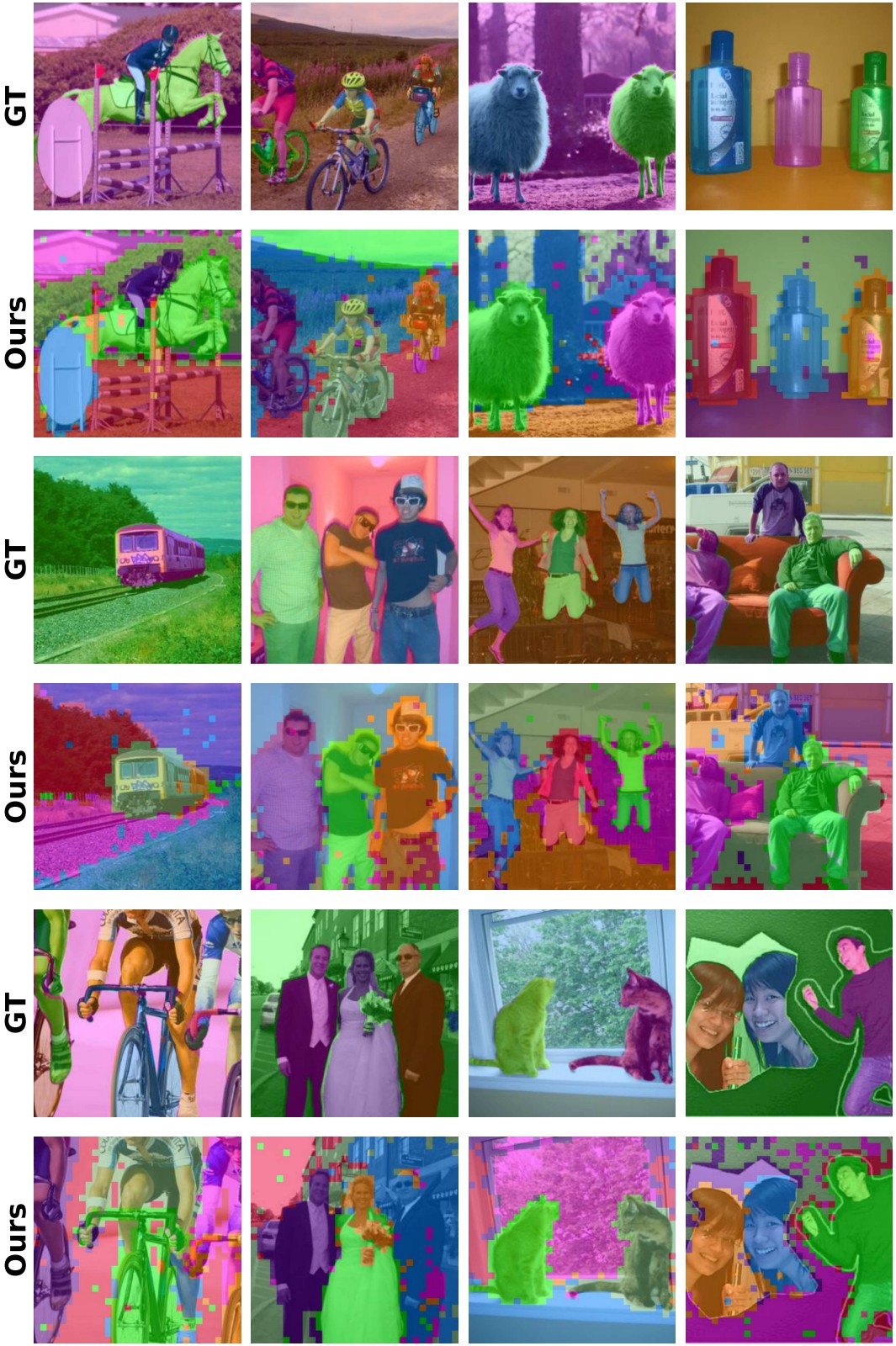

Figure 8: **Unsupervised Object Segmentation.** We show visualizations of predicted segments on VOC.

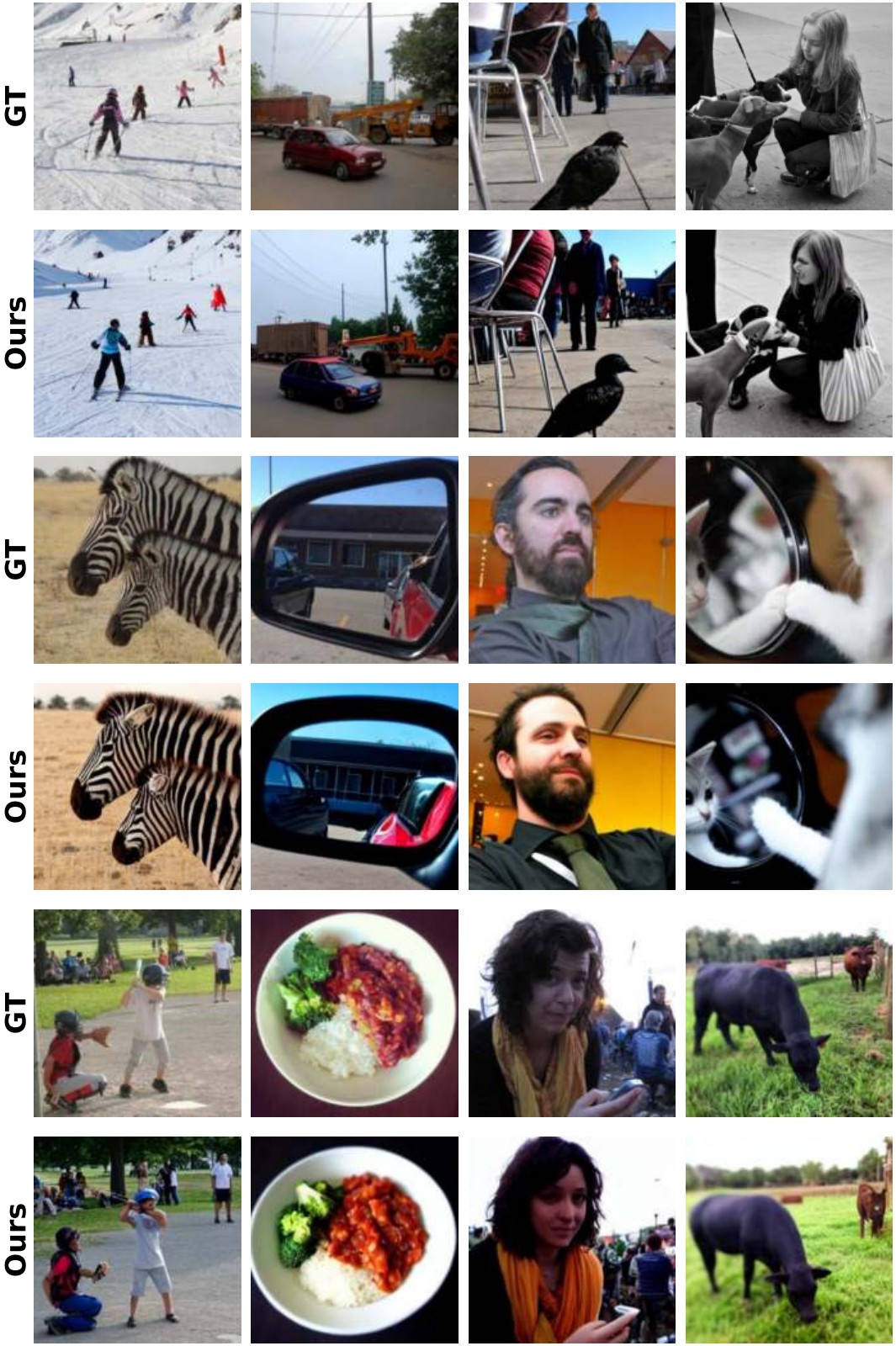

Figure 9: **Generation Results.** We show sample images from COCO, reconstructed by SlotAdapt conditioned on slots.

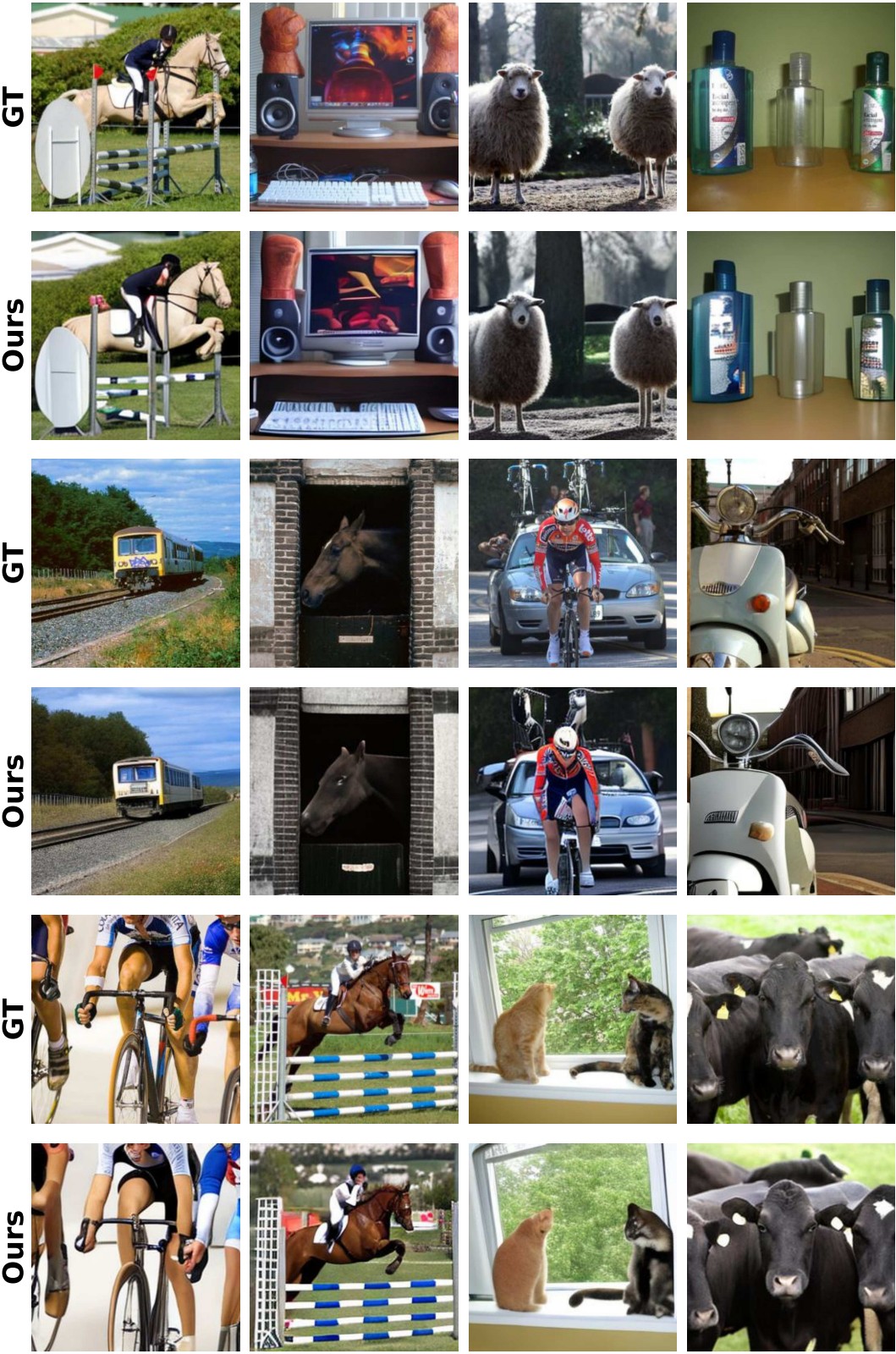

Figure 10: **Generation Results.** We show sample images from VOC, reconstructed by SlotAdapt conditioned on slots.

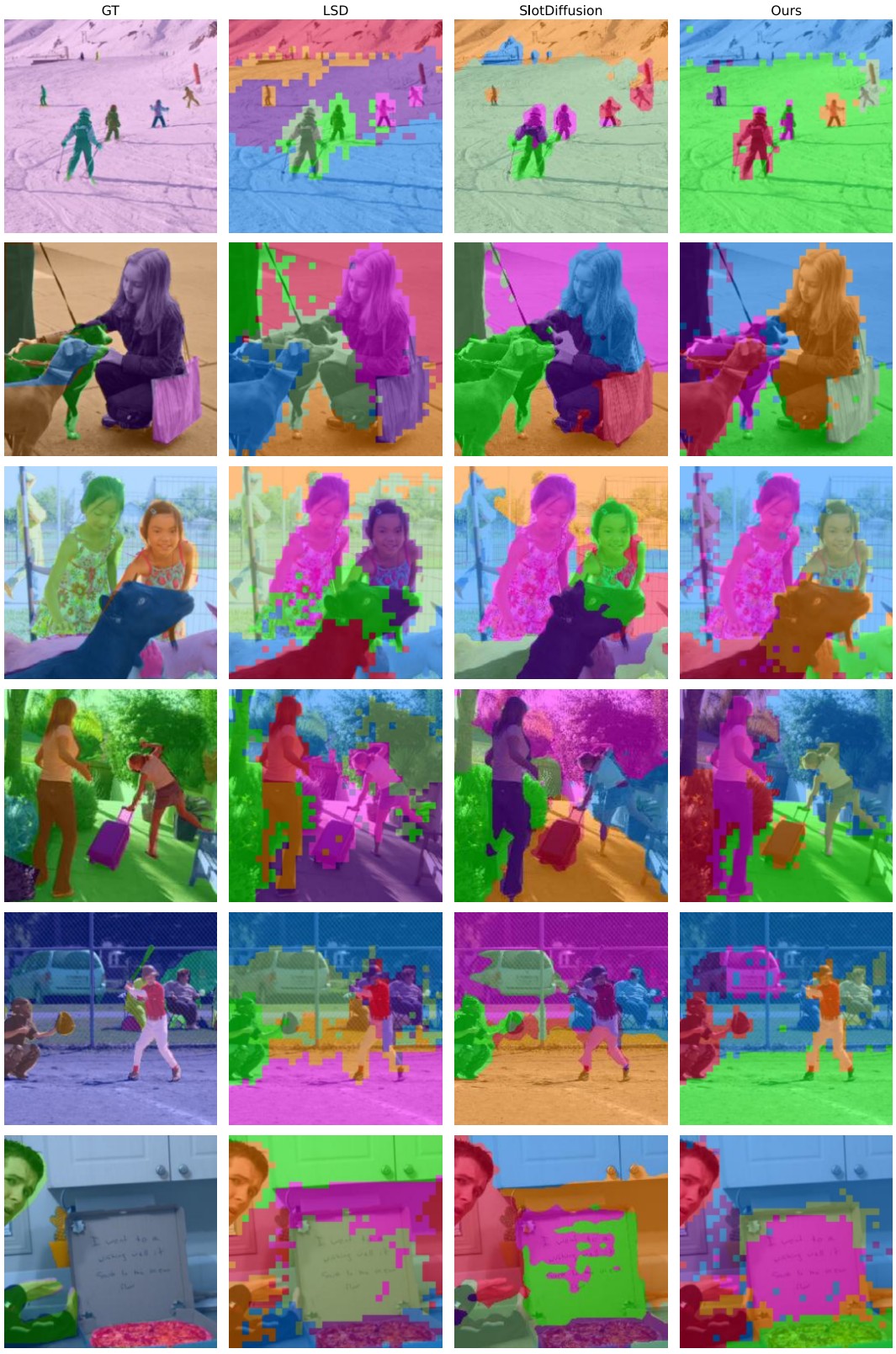

Figure 11: **Segmentation comparisons with other methods** We show visualizations of predicted segments on COCO dataset. Compared to other models, our model tends to produce more coherent masks with fewer fragmented segments.

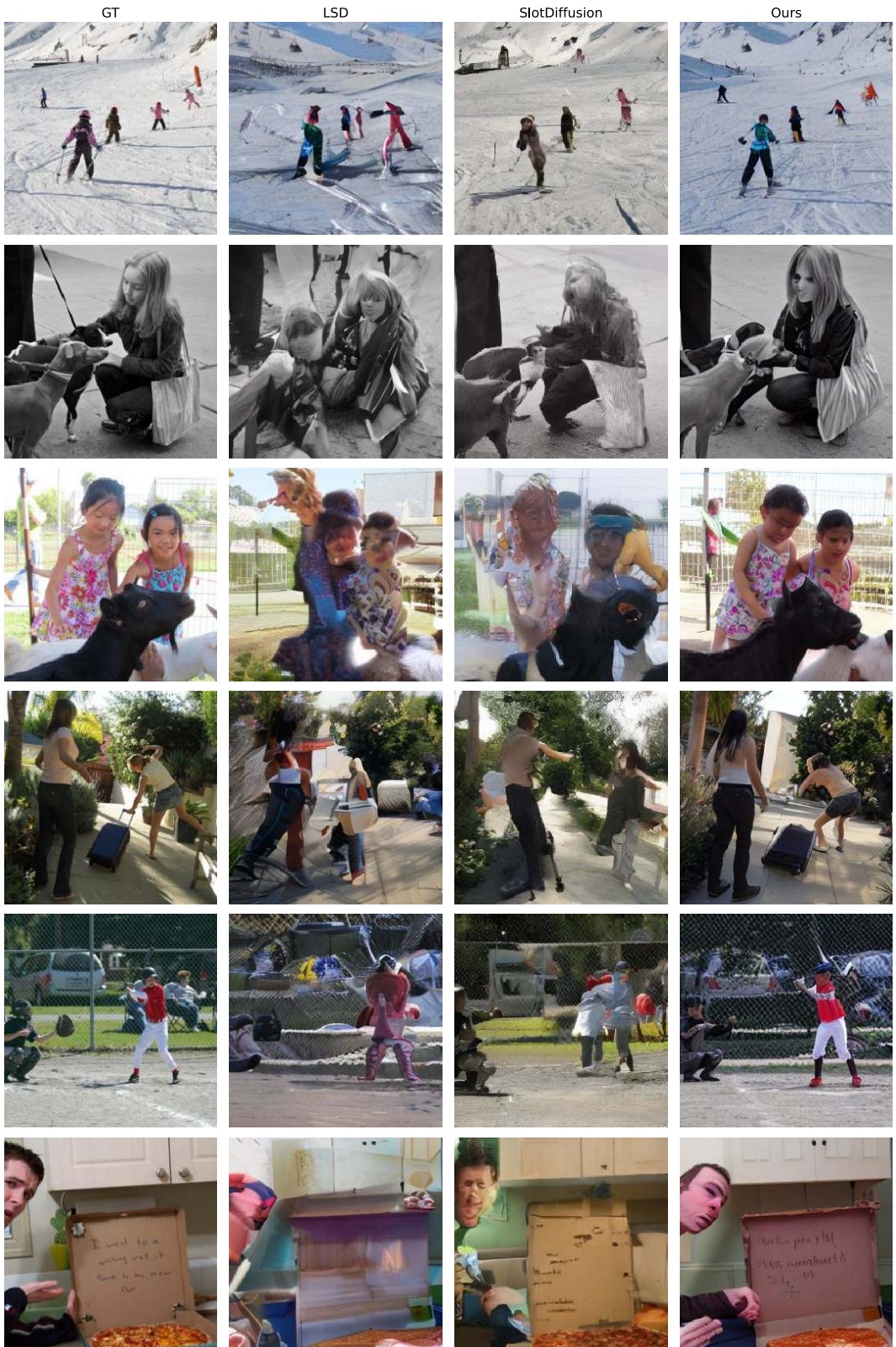

Figure 12: **Generation comparisons with other methods.** We show visualizations of generated images on COCO dataset. Compared to other models, our model can generate better reconstructions.

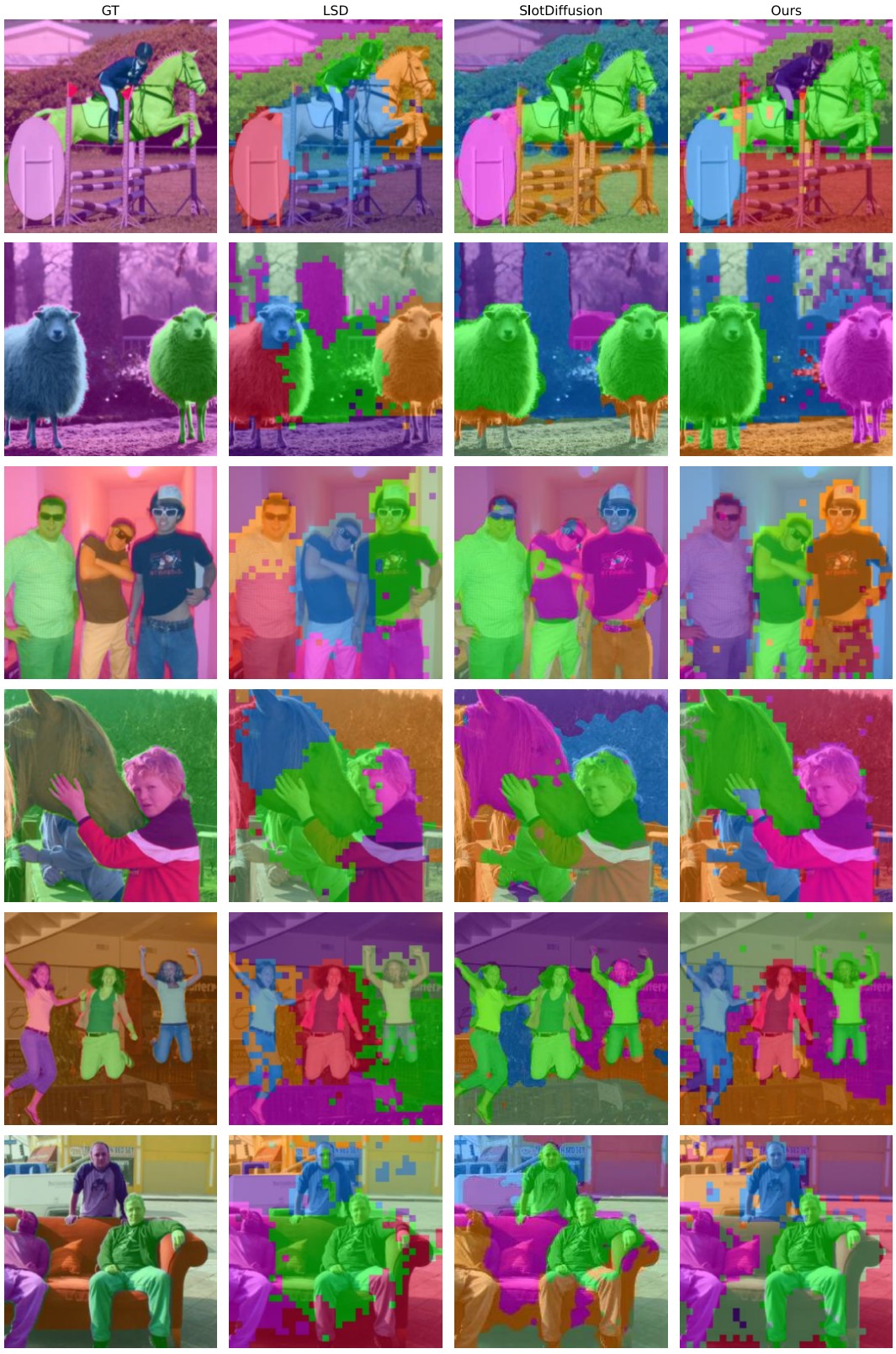

Figure 13: **Segmentation comparisons with other methods.** We show visualizations of predicted segments on VOC dataset. Compared to other models, our model tends to produce more coherent masks with fewer fragmented segments.

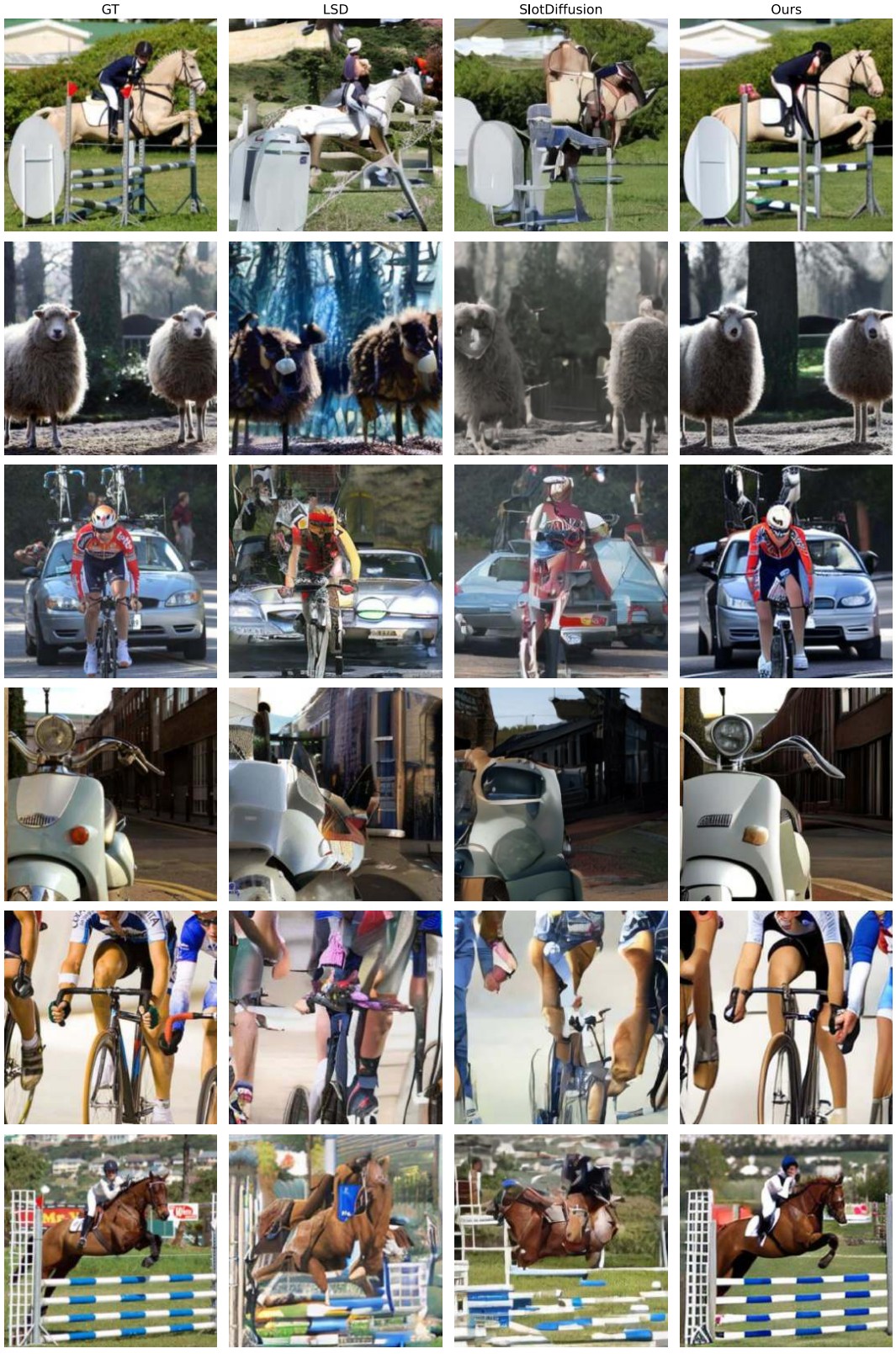

Figure 14: **Generation comparisons with other methods.** We show visualizations of generated images on VOC dataset. Compared to other models, our model can generate better reconstructions.

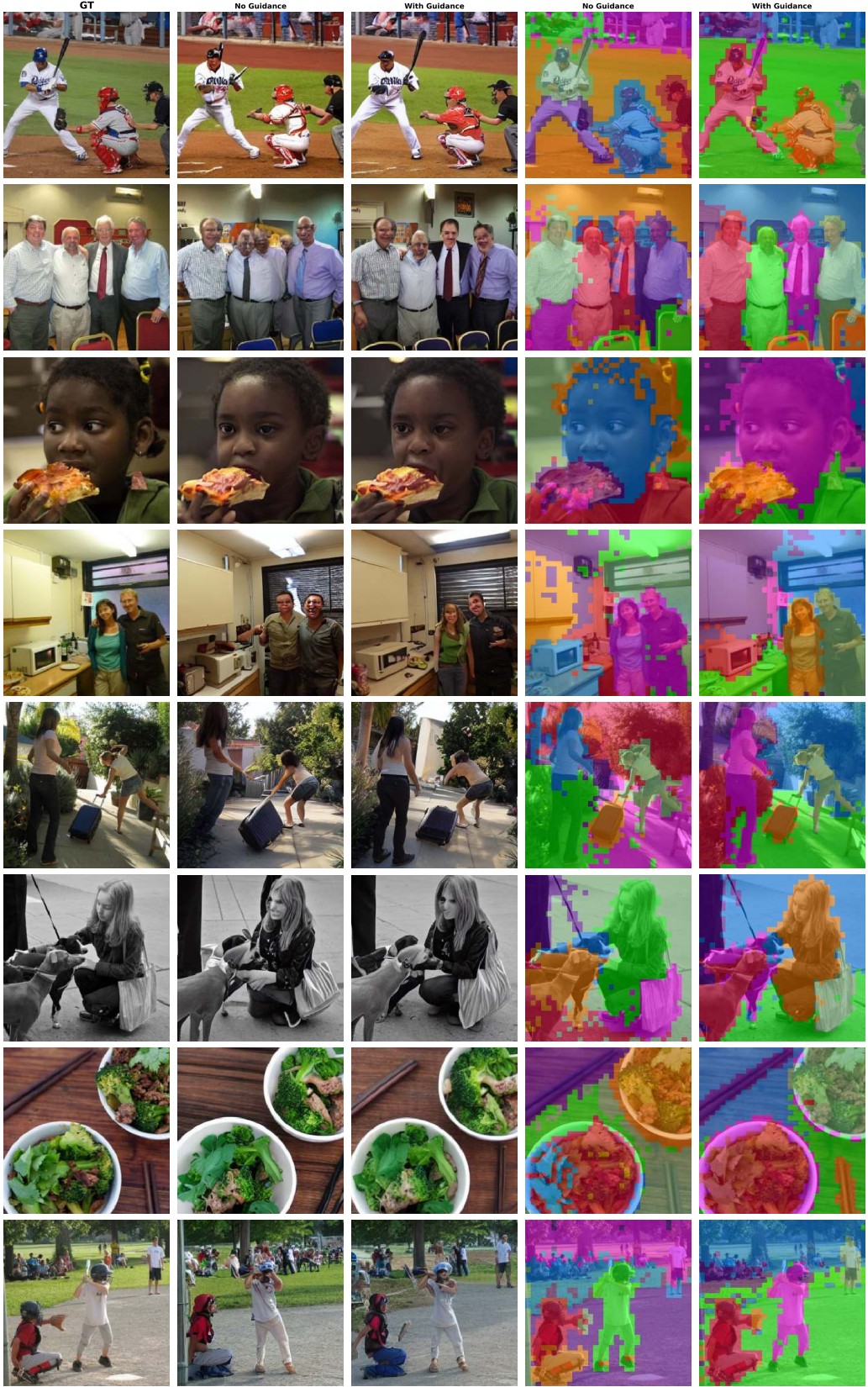

Figure 15: **Qualitative comparison: with vs. without guidance.** We visualize generated images and predicted segments on COCO dataset. Guidance helps to generate better aligned objects and to mitigate the "part-whole" hierarchy problem in object segmentation task.

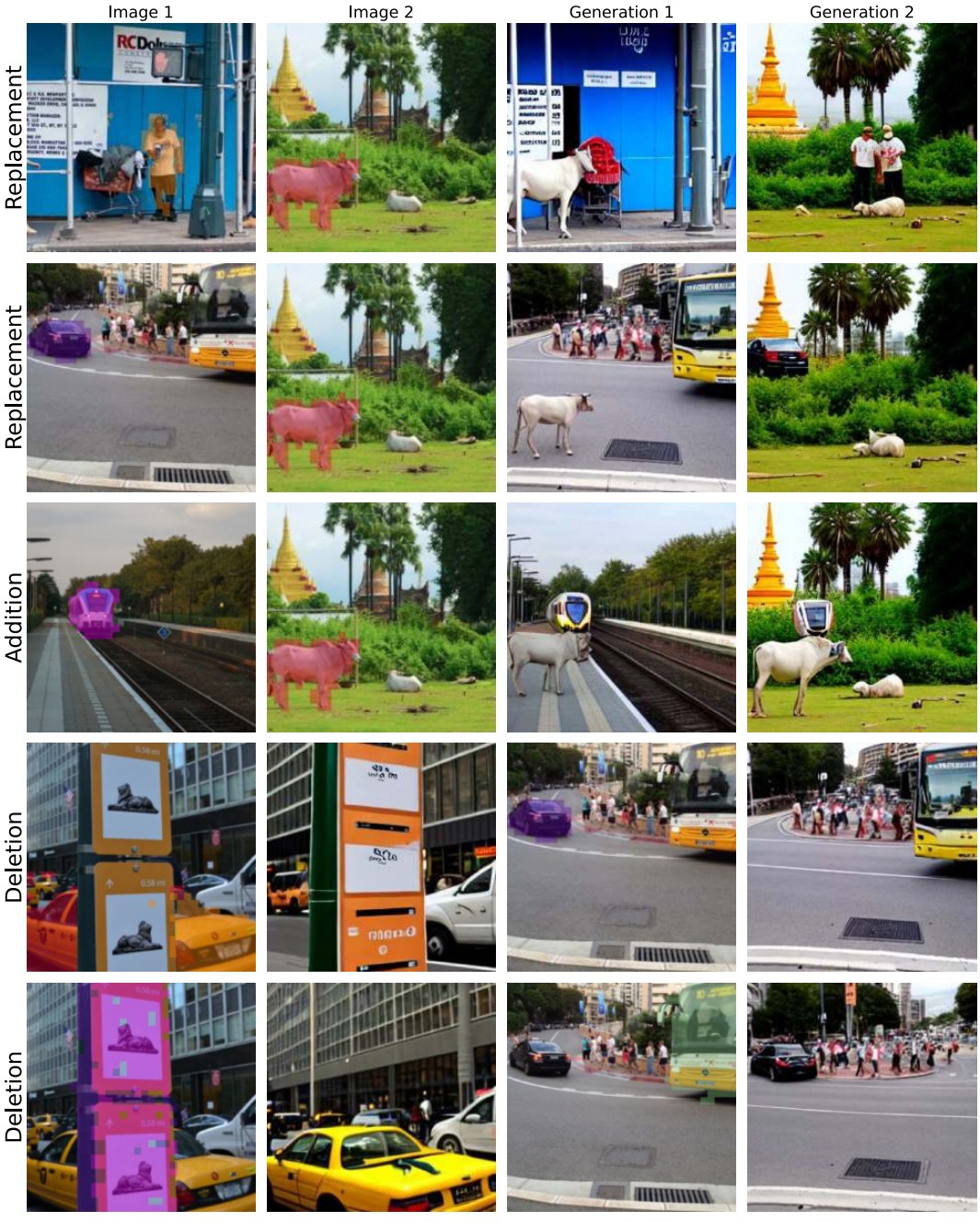

Figure 16: **Compositional Generations and Editing.** We show visualizations of generated images when the slots are manipulated. In the first two rows, we exchange the highlighted slots. In the middle row, we simply add the highlighted slot to the other image. In the last two rows, we delete the highlighted slots to manipulate the image. For all scenarios, our model can successfully generate realistic images in complex scenarios. This shows our model's ability for compositional generation and editing.

A.4  COMPARISON WITH GLASS AND SPOT

This section presents a comprehensive comparison between our method and the concurrent work GLASS (Singh et al., 2024) and SPOT (Kakogeorgiou et al., 2024), highlighting both quantitative and qualitative differences.

**Methodological Distinctions:**  GLASS leverages extra information, such as class labels or image captions, to enhance its performance. While this approach yields certain advantages, it introduces limitations. Primarily, GLASS struggles to differentiate between instances of the same class due to its reliance on semantic masks as pseudo ground truth. Additionally, the need for additional information restricts the applicability of GLASS in fully unsupervised scenarios. In contrast, our method operates without any external supervision, successfully segments individual instances even within the same class, and captures nuanced scene information without relying on pre-defined semantic categories. We should also note that GLASS does not present any compositional editing results.

SPOT operates in a fully unsupervised manner by introducing (i) an attention-based self-training mechanism that distills improved slot-based attention masks from the decoder to the encoder, and (ii) a patch-order permutation strategy for autoregressive transformers to better utilize slot representations during reconstruction. Although SPOT achieves strong object segmentation performance, particularly on complex real-world images, it still faces challenges in accurately differentiating fine-grained object instances within the same class. We should also note that SPOT does not present any compositional editing or reconstruction results.

Figure 17 provides visual comparisons on both COCO and VOC datasets. Our model demonstrates superior performance in instance differentiation, accurately segmenting multiple instances of the same class (as shown in rows 1, 2, and 4). Furthermore, it excels in scene understanding, capturing meaningful elements such as trees and house roofs (row 3) without explicit labeling.

Table 11 presents a quantitative comparison of our method with GLASS and SPOT. In terms of semantic overlap, GLASS excels, which is attributable to its use of semantic masks for supervision. However, on complex datasets like COCO, which features multiple instances per image, SlotAdapt achieves comparable performance without any additional supervision. SPOT, while fully unsupervised, focuses on enhancing slot representations through self-training and patch-order permutation strategies. Although it performs well in capturing global scene structures, particularly on COCO, it lags behind our method in fine-grained instance differentiation, as reflected in the FG-ARI scores.

These results underscore the robustness and versatility of our unsupervised approach, particularly in handling complex, multi-instance scenarios. Our method demonstrates that high-quality object segmentation can be achieved without relying on external supervision, offering a more flexible and generalizable solution for diverse image understanding tasks.

Table 11:  Unsupervised object segmentation comparisons with the concurrent work GLASS and SPOT on VOC (left) and COCO (right). We would like to point out that both GLASS and GLASS[†] use extra supervision such as class labels or image caption. SPOT is a fully unsupervised method.

| PASCAL VOC | FG-ARI | mBO$^i$ | mBO$^c$ | MS COCO | FG-ARI | mBO$^i$ | mBO$^c$ |
|---|---|---|---|---|---|---|---|
| Ours | 28.8 | 51.6 | 52.0 | Ours | **42.3** | 31.5 | 34.8 |
| Ours + Guidance | **29.6** | 51.5 | 51.9 | Ours + Guidance | 41.4 | 35.1 | 39.2 |
| GLASS[†] | — | **60.4** | **68.4** | GLASS[†] | — | 34.3 | 45.2 |
| GLASS | — | 58.1 | 36.1 | GLASS | — | **35.3** | **46.3** |
| SPOT w/o ENS | 19.7 | 48.1 | 55.3 | SPOT w/o ENS | 37.8 | 34.7 | 44.3 |
| SPOT w/ ENS | 19.9 | 48.3 | 55.6 | SPOT w/ ENS | 37.8 | 35.0 | 44.7 |

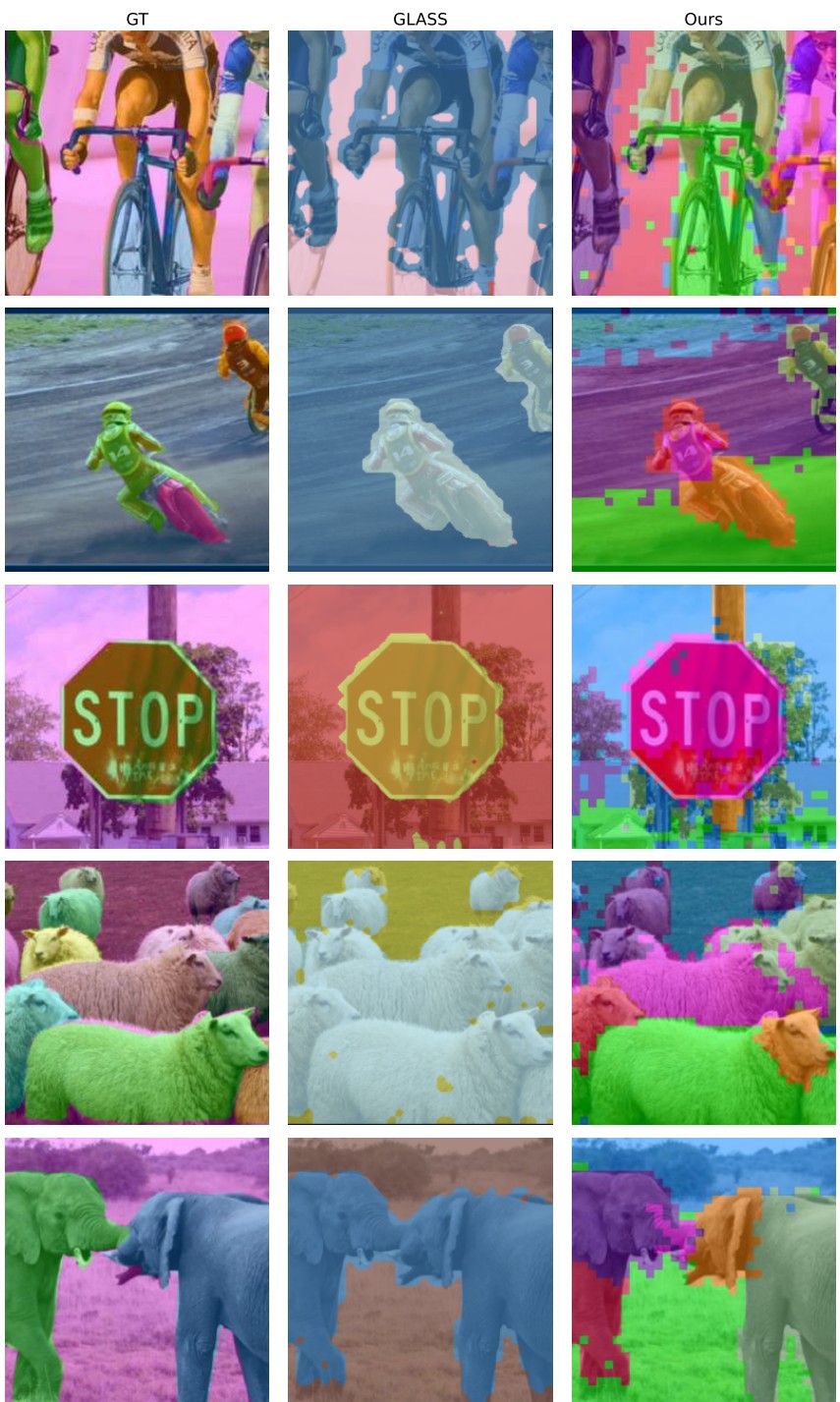

Figure 17: **Unsupervised Object Segmentation.** We show visualizations of predicted segments for SlotAdapt vs. GLASS on real world datasets (VOC and COCO).

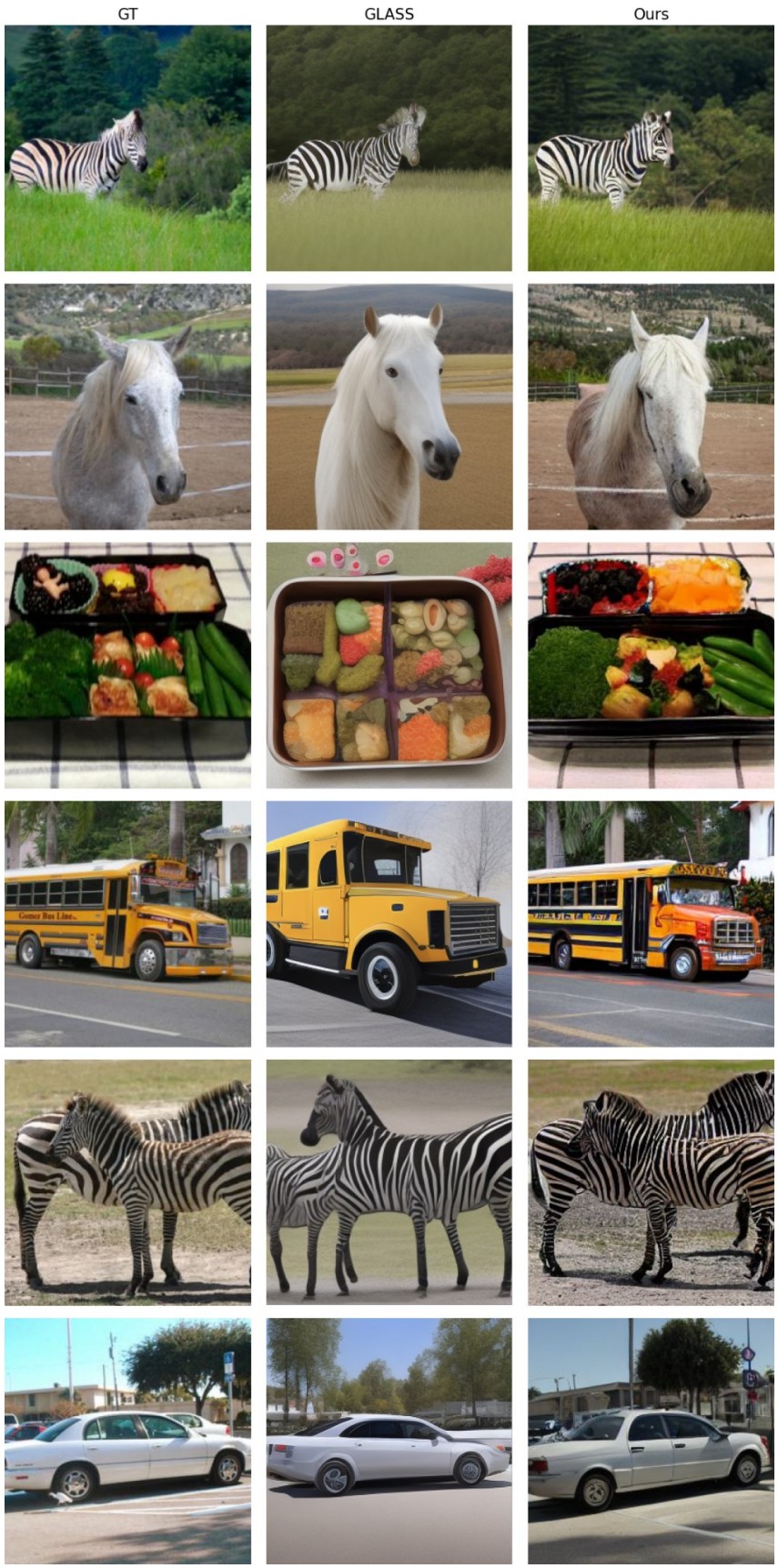

Figure 18: **Generation results.** We show visualizations of generated images by SlotAdapt vs. GLASS on COCO and VOC.