# OpenReview forum: "Slot-Guided Adaptation of Pre-trained Diffusion Models for Object-Centric Learning and Compositional Generation"
_ICLR.cc/2025/Conference — ICLR 2025 Poster_

### Official Review · Reviewer_qbBW · 2024-10-19

**Soundness:** 3
**Presentation:** 3
**Contribution:** 3
**Rating:** 8
**Confidence:** 4

**Summary:**

This paper proposes SlotAdapt, a novel approach to leverage large-scale pre-trained diffusion models (DMs) for the task of object-centric learning (OCL). There are priors work trying to combine OCL with DMs, yet they usually just treat slots as text embeddings, and directly input them to the cross-attention layer in the diffusion U-Net. SlotAdapt instead inserts a new adapter layer to the U-Net to condition on slots. With this design, slots do not need to lie in the text embedding space, leading to better object-centric representations. Authors conduct extensive experiments including both segmentation and generation on real-world datasets COCO and VOC. The results are impressive and the generation result is clearly better than previous works.

**Strengths:**

- The adapter design is reasonable, and the motivation behind it is clear -- slots capture object semantic information, which might be incompatible with the text embedding space. With this new adapter layer, one can freeze the pre-trained DM while not sacrificing the expressiveness of learned object slots.
- The guidance loss is novel. While there are concurrent work (e.g. GLASS) applying a similar loss, they usually require external supervisions such as captioner (to get text from input images). I am surprised that just letting slot attention mask and cross-attention mask to guide each other can work so well (I would expect some stop gradient operation to stabilize training). The ablation studies in Table 2 are comprehensive and answer my concerns very well.
- As far as I know, SlotAdapt is the first work showing compositional generation ability on real-world images (prior work only show segmentation results). I find the results in Fig.6 impressive.

**Weaknesses:**

I do not find any big weakness of this paper. There is one claim that sounds not precise to me and I hope the author can revise it.
- In line 468 and Fig.5 caption, authors claim that SlotAdapt "generates nearly perfect reconstructions". I do not think this is the case, as apparently the reconstructed images are clearly different from the input image. This result itself is not a weakness of the method, as slots are compressed representations and cannot preserve all visual details. However, I do believe the statement itself is wrong.

Minor:
- Please unify the name of U-Net in the paper. In Fig.1 and line 230 it is called "Unet", in line 207 "U-Net", in line 223 and 224 "UNet".
- Line 320 "v1.5 for MOVi-E; and COCO", I think this ";" is redundant.
- The generation results of SlotAdapt are impressive, yet authors only put small figures in the main paper (Fig. 5 and 6). I understand this is due to page limit. Please add text to refer readers to large figures in Appendix Fig. 8 and 10.

**Questions:**

1. What will happen if you add the guidance loss from the beginning of training? Will the training diverge?
2. Why using SD 1.5 on COCO and SD 2.1 on VOC? In SlotDiffusion Appeidix E.5, they do mention that using the v-prediction objective sometimes leads to degenerate results. Is it also the case here as SD 2.1 is using v-prediction?
3. Have you tried classifier-free guidance (CFG)? I am curious if using a larger CFG scale, will the reconstruction result be even better.
4. Instead of pooling all slots to form the register token, I guess an interesting alternative is 1) add an additional slot in your slot attention module, and 2) only input this new slot to the cross-attention layer. This slot is dedicated to capture global information and may perform better than pooling object slots.

---

> ### Author Response · Authors · 2024-11-20
> **Response to reviewer qbBW**
>
> We thank reviewer qbBW for their detailed comments and valuable feedback. The reviewer found our motivation for using adapters reasonable, and acknowledged the novelty of both our guidance loss and compositional generation ability, particularly noting our results on complex real-world images. They expressed concern about our language regarding reconstructions. We agree that there are some changes in the images (also noted by reviewers 8w5Y and Hqxz) and will revise this language in the updated version. We will also address their noted minor issues regarding language/grammar errors and missing text references for figures.
>
> The reviewer raised four questions, which we address below:
> 1. Effects of adding guidance loss from the beginning of training:
> Our intuition was that initially, with random initialization, segmentation masks would lack useful supervisory information. Therefore, we chose to introduce guidance loss after the model learned useful image details. However, we conducted the suggested experiment. While training didn't diverge in this experiment, results were inferior to our original strategy of introducing guidance loss after 40K iterations:
>
> | | FG-ARI| mBO-i | mIoU-i | mBO-c | mIoU-c |
> |---------|---------|-----------|---------|-----------|---------|
> | Start from 0 | 37.85 | 32.65 | 33.99 | 36.059 | 39.254 |
> | Original (start after 40K) | 41.4 | 35.1 | 36.1 | 39.2 | 41.4 |
>
>
> 2. Choice of SD 1.5 for COCO and SD 2.1 for VOC:
> In initial experiments, SD1.5 performed better on COCO while SD2.1 excelled on VOC. We attribute this to dataset size differences: COCO has approximately 118K images while VOC has 12K. Given SD2.1's enhanced capabilities and larger training dataset, we believe it adapted more easily to VOC's relatively limited dataset. Regarding v-prediction, we didn't observe the degeneration issues mentioned in SlotDiffusion's Appendix E.5, likely because we used pretrained models rather than training from scratch.
>
> 3. Classifier-free guidance (CFG) experimentation:
> We initially used a CFG value of 2.5 based on observed visual improvements. SlotDiffusion tried CFG but saw no improvements while LSD uses CFG value of 1.3. Following the reviewer's question, we conducted comprehensive CFG experiments with our best model on COCO dataset:
>
> | CFG Value | FID | KID*1000 |
> |-----------|-----|----------|
> | 7.5 | 21.350 | 6.271 |
> | 5.0 | 17.558 | 4.236 |
> | 2.5 | 13.734 | 2.151 |
> | 2.0 | 12.427 | 1.424 |
> | 1.5 | 11.041 | 0.590 |
> | 1.4 | 10.880 | 0.459 |
> | 1.3 | 10.857 | 0.388 |
> | 1.2 | 11.057 | 0.492 |
>
> We will revise compositional generation and reconstruction results presented in the paper accordingly. Thanks for the suggestion.
>
> 4. Alternative register token approach:
>
> We appreciate the reviewer's suggestion of using an additional slot in the slot attention module dedicated to capturing global information. We implemented the suggestion and our experiments showed mixed results:
>
> | | FG-ARI | mBO-i | mIoU-i | mBO-c | mIoU-c | FID | KID |
> |---------|---------|-----------|---------|-----------|---------|-----|-----|
> | Additional Slot Token | 43.83 | 31.9 | 32.4 | 35.5 | 37.3 | 11.212 | 0.431 |
> | Slot Average Token | 42.3 | 31.5 | 34.8 | 34.8 | 38.5 | 10.857 | 0.388 |
>
> While this approach improved several segmentation metrics, reconstruction performance slightly decreased. We hypothesize that the additional slot token's limited representation capacity (due to competitive attention in slot attention) prevents it from fully capturing global scene information, unlike our slot average token which incorporates information from all tokens. We will include these findings in the updated version.

---

> ### Comment · Reviewer_qbBW · 2024-11-20
> **Re: rebuttal**
>
> I thank the authors for their reply. After reading other reviewers' comments, I will maintain my original rating of accept. I believe this work is an important milestone for the object-centric representation field.

---

> > ### Author Response · Authors · 2024-11-29
> > **Official Comment by Authors**
> >
> > We thank the reviewer for their positive assessment and constructive suggestions to strengthen the paper. With the rebuttal period nearing its deadline and the reviews being finalized, we want to assure the reviewer that we are ready to address any further concerns or questions that the reviewer may have.

---

### Official Review · Reviewer_8w5Y · 2024-10-29

**Soundness:** 2
**Presentation:** 4
**Contribution:** 2
**Rating:** 6
**Confidence:** 4

**Summary:**

This paper introduces an object-centric learning approach, SlotAdapt, with the focus on scaling to real-world dataset. Instead of directly frozening or fine-tuning the pretrained stable diffusion model, SlotAdapt adds an adapter module on top of pretrained stable diffusion so that the learned slots are not strictly limited to lie in the text embedding space, enabling better slot-object alignment for real world images. The resulted flexibility of slots is traded with an additional module, Adapter, needing to be trained when compared with the frozen pretrained stable diffusion case. SlotAdapt also self-supervises slot attention maps with cross-attention maps for semantic alignment. Experiment results show improvements on the segmentation performance.

The main contribution of this paper comes from the architecture design.

**Strengths:**

1. The writing is smooth and easy to follow

2. The paper is tackling an important problem in the object-centric learning community, i.e., pushing the boundary of object-centric learning approaches towards real world images, and shows some non-trival improvements on the segmentation performance.

3. The proposed approach is straightforward and effective by borrowing and combining ideas from LSD and T2I adapters without requiring additional supervision. It largely benefits from the prior knowledge stored in pretrained stable diffusion models while allows flexible representation learning ability of slots.

**Weaknesses:**

1. This paper claims, "Our method performs remarkably well on complex real-world images for compositional generation, in contrast to other slot-based generative methods in the literature." However, only qualitative results are provided, even without a comparison with existing approaches. It should be noted that existing approaches like LSD [1] and Slate [5] can already perform compositional generation on real world datasets like FFHQ in Figures 3, 4, and 10 https://arxiv.org/pdf/2303.10834. Comparing your approach with LSD and/or other approaches on Movi-E or COCO is necessary to support your claim. Otherwise, the sentence should be revised. Similarly, quantitative scores like FID are helpful to demonstrate the statements on high fidelity generation in the section "Generation and Compositional Editing".

2. For the guidance strategy, the FG-ARI in Table 2 shows that guidance doesn't help a lot for segmentation, while Fig 2 qualitatively illustrates that guidance might help a bit. Are the qualitative results of Fig 2 possibly due to random seeds used in the experiments? Is there any intuition that guidance can help avoid over-splitting? After all, there is no external supervision, it is possible that both slot attention map and cross attention map oversplit. Furthermore, in section 4.1, it says "In turn, the generated (reconstructed) images are more faithful to the original input images", which is not that persuasive depending on what you focus on (the clothes of base ball player, the closets and ceiling lamp).


3. There are several overclaims in the paper. For example, in Figure 5, it says "SlotAdapt generates nearly perfect reconstructions" while the figure shows that details are still largely different between GT and reconstruction. In Abstract and Conclusion, it claims outperforming baselines in compositional generation; however, there is no qualitative and quantitative comparison in any section in compositional generation.

4. Now that the paper is titled with Compositional Generation, some relevant reference, such as [2][3][4] etc., are suggested to be discussed in related works though not directly compared.

5. Limitation discussion. (1)  LSD paper finds that their approach shows significant gains in complex naturalistic scenes but a somewhat diminished performance in terms of segmentation and representation when the dataset consists of only visually simple and monotonous scene images like CLEVR. Does SlotAdapt share similar limitation when it comes to simple scenes like CLEVR or CLEVRTex? (2) Although adopting pretrained stable diffusion models allows using prior knowledge to improve segmentation performance on real-world images like COCO,  it on the other hand might limit the model's ability to discover per-dataset components, for example facial slots in LSD [1].


[1] Jiang, "Object-Centric Slot Diffusion", NeurIPS 2023

[2] Jiang, " Generative neurosymbolic machines", NeurIPS 2020

[3] Wang, "Slot-VAE: Object-Centric Scene Generation with Slot Attention", ICML 2023

[4] Wu, "NEURAL LANGUAGE OF THOUGHT MODELS", ICLR 2024

[5 Singh, "Illiterate DALL-E Learns to Compose", ICLR 2022

**Questions:**

1. Does the number of slots used in the experiments have an impact on the part-whole results? For COCO dataset, do you simply assume the slot number is 80? In that case, is the training slow?

2. Looks like the quantitative scores of baselines in Table 3 are directly copied from LSD paper. In that case, it should be guaranteed that the experimental settings of yours and that of LSD paper are identical for fair comparison.

---

> ### Author Response · Authors · 2024-11-20
> **Response to reviewer 8w5Y**
>
> We thank reviewer 8w5Y for their in-depth analysis and valuable feedback. The reviewer found our paper easy to follow and our contributions effective and novel. They raised several concerns, which we address below:
>
> Regarding the evaluation of reconstruction performance and compositional generation ability:
> Following their suggestion, we evaluated all models for reconstruction performance on COCO using FID and KID scores, which measure the similarity between the generation distribution and ground truth data distribution. We also assessed the compositional generation ability of our model, in comparison to LSD and SlotDiffusion, by applying SlotDiffusion's compositional generation evaluation procedure, where slots in the batch are randomly mixed and generation is conditioned on these mixed slots (see SlotDiffusion paper Fig. 6; implementation from the official SlotDiffusion repository: https://github.com/Wuziyi616/SlotDiffusion/blob/main/slotdiffusion/img_based/test_comp_gen.py). Below are the results for both compositional generation and reconstruction performance:
>
> **Compositional Generation**
>
> | Method | FID | KID*1000 |
> |----|-----|------|
> | Ours w/ guidance | 40.568 | 34.381 |
> | LSD | 167.232 | 103.482 |
> | SlotDiff | 64.213 | 57.309 |
>
> **Reconstructions**
>
> While we initially used CFG=2.5 for all generations, we discovered that CFG=1.3 performs better and include those results below (CFG parameter search was conducted following reviewer qbBW's question):
>
> | Method | FID | KID*1000 |
> |----|-----|------|
> | Ours CFG=1.3 | 10.857 | 0.388 |
> | Ours CFG=2.5 | 13.734 | 2.151 |
> | LSD | 35.537 | 19.086 |
> | SlotDiff | 19.448 | 5.852 |
>
> 2. Our motivation for using guidance loss is to leverage the generative prior of pretrained diffusion models. This prior reflects learned internal representations of objects' identity, structure, and composition, derived from training on vast datasets. We anticipate this approach helps mitigate over and under-segmentation problems, as evidenced in Table 2 (Guidance ablations). While we don't see improvement in FG-ARI scores for object discovery, the improvements in mBO and mIoU metrics are substantial, indicating better spatial accuracy in segmentation results.
>
>      We also verified that the results in Fig. 2 are not seed-dependent. Changing seeds produces the same outcome, with objects simply bound to different slots. We will include these results in the updated version.
>
> 3. We acknowledge the incorrect language regarding "perfect" reconstructions (also noted by reviewers qbBW and uBXM) and will revise it to "highly faithful to the original images." We evaluated all methods using FID and KID and the results show that our method outperforms all baselines in both reconstruction and compositional generation tasks. We will include these results and additional qualitative examples in the updated paper.
>
> 4. We appreciate the reviewer highlighting missing references, which we will add in the updated version.
>
> 5. The reviewer raised an important point about limitations. We agree that using a pre-trained diffusion model, primarily trained on real-world data, might limit performance on synthetic datasets like CLEVR/CLEVRTex. While facial information might be easier to adapt due to the presence of face data in the training set of diffusion models, synthetic datasets could pose greater challenges. We will discuss this limitation in the updated paper.
>
> Regarding the reviewer's two questions:
>
> 1. Number of slots and usage of 80 slots:
> We determine the slot count by the maximum number of objects per image, following DINOSAUR's approach, which all baselines (LSD and Slot Diffusion) also adopt. Slots represent instances rather than classes, allowing different instances of the same class to occupy different slots. Nevertheless, following the reviewer's question, we tested using 80 slots (matching COCO dataset classes). This resulted in over-splitting and decreased performance due to the slot count far exceeding typical scene object counts. Training with 80 slots is indeed slower due to increased attention queries in slot attention (80 vs. 7) and adapter keys (80 vs. 7). Rather than using the class count for slot numbers, a better alternative would be dynamically setting slots per image - an active research topic [1] as we mention in our conclusions.
>
>
> ||FG-ARI | mBO-i | mIoU-i | mBO-c | mIoU-c | FID | KID |
> |---------|---------|-----------|---------|-----------|---------|-----|-----|
> | 80 slots | 18.1 | 23.2 | 26.4 | 28.6 | 32.9 | 114.236 | 64.610 |
> | 7 slots | 41.4 | 35.1 | 36.1 | 39.2 | 41.4 | 10.857 | 0.388 |
>
> 2. Results on MOVi-E dataset and experimental settings (Table 3):
> The MOVi-E results come from the LSD paper, with identical experimental settings except for our use of pretrained diffusion model and VAE.
>
> [1] Fan, K., Bai, Z., Xiao, T., He, T., Horn, M., Fu, Y., ... & Zhang, Z. (2024). Adaptive slot attention: Object discovery with dynamic slot number. CVPR 2024.

---

> > ### Comment · Reviewer_8w5Y · 2024-11-26
> >
> > Thanks for the response. The added experiments are helpful to support the claims in the paper.

---

> > > ### Author Response · Authors · 2024-11-29
> > > **Official Comment by Authors**
> > >
> > > We thank the reviewer for their positive assessment and constructive suggestions to strengthen the paper. With the rebuttal period nearing its deadline and the reviews being finalized, we want to assure the reviewer that we are ready to address any further concerns or questions that the reviewer may have.

---

### Official Review · Reviewer_Hqxz · 2024-11-04

**Soundness:** 2
**Presentation:** 2
**Contribution:** 2
**Rating:** 5
**Confidence:** 4

**Summary:**

The paper introduces SlotAdapt, an object-centric learning method that integrates slot attention with pre-trained diffusion models using adapter layers for slot-based conditioning. This work can significantly avoid biases generated from the text-centric conditioning models. The proposed adapter layers with slot attention can enhance the model's alignment with image objects without external supervision.

**Strengths:**

Introducing adapter layers for slot-based conditioning enables the model to deviate from text embedding space representations while preserving the generative capabilities of pre-trained diffusion models.

The compositional image generation results on complex real-world images demonstrate the effectiveness of the model in handling compositional generation tasks.

**Weaknesses:**

The paper inconsistently uses both "UNet" and "Unet" to refer to the U-Net architecture. It is advised that the authors adhere to the standard capitalization "UNet" consistently across the document. Additionally, it is recommended to introduce the term "register token" earlier in the paper to establish its significance and reducing confusion. It is crucial for the authors to perform a thorough review and correct grammar and syntax problems.
The reconstructed images may exhibit slight changes or artifacts comparing to the source image. This issue might be attributed to over-segmentation, which could lead to the misrepresentation of certain image components during the generation process.

**Questions:**

How can the authors demonstrate that the unique contributions of their model extend beyond the use of adapters, which are becoming increasingly common in the LDM？
The paper's outcomes appear heavily reliant on segmentation quality. Could the authors clarify if the model's success is primarily due to the performance of segmentation, and whether integrating a superior segmentation method could achieve similar results?

---

> ### Author Response · Authors · 2024-11-20
> **Response to reviewer Hqxz**
>
> We thank reviewer Hqxz for their detailed comments and valuable feedback..The reviewer highlights our introduction of adapter layers for slot-based conditioning as a key strength, emphasizing that it allows the model to move beyond text embedding space representations while maintaining the generative capabilities of pre-trained diffusion models. They also acknowledge our results for compositional image generation on complex real-world images such as COCO. Their concerns primarily focus on writing and grammar issues, as well as slight changes and artifacts in the reconstructed images. We will address all grammatical and syntax issues in the updated version of the paper. Regarding reconstruction quality, we note that inversion methods are generally approximate, and reconstructions obtained by pretrained diffusion models are inherently imperfect; indeed, exact inversion remains an active research topic in the field [1,2]. We have also evaluated our method's generation quality using FID and KID metrics (as suggested by reviewer 8w5Y) and compared them to the baselines (see the reply to the reviewer 8w5Y).
>
> The reviewer raised two key questions. First, regarding contributions beyond the usage of adapters: our method also introduces register token and guidance loss, which significantly improve results. We present these improvements in Table 1 and Table 2 of the main paper, respectively, demonstrating that both guidance loss and register token enhance performance.
>
> Second, regarding whether our method's performance relies on segmentation quality: our primary goal is to learn object-centric representations rather than merely performing segmentation. We then feed these object-centric representations to the diffusion model to guide the generation process. Our method can both learn representations specific to scene objects and perform object segmentation. We chose slot attention for learning object-centric representations, following LSD and SlotDiffusion. To explore potential improvements, we tested BOQ-SA [3], an enhanced version of slot attention which also yields better segmentation results as asked by the reviewer. The results show modest improvements across several metrics, as detailed below:
>
> | Method | FG-ARI | mBO-i | mIoU-i | mBO-c | mIoU-c |
> |--------|---------|--------|---------|--------|---------|
> | Slot Attention | 42.3 | 31.5 | 31.7 | 34.8 | 38.5 |
> | BOQ-SA | 42.2 | 31.2 | 32.551 | 35.266 | 37.82 |
>
> [1] Wallace, B., Gokul, A., & Naik, N. (2023). EDICT: Exact Diffusion Inversion via Coupled Transformations. In CVPR (pp. 22532-22541).
>
> [2] Hong, S., Lee, K., Jeon, S. Y., Bae, H., & Chun, S. Y. (2024). On Exact Inversion of DPM-Solvers. In CVPR 2024.
>
> [3] Jia, B., Liu, Y., & Huang, S. (2023). Improving object-centric learning with query optimization. ICLR 2023.

---

> > ### Comment · Reviewer_Hqxz · 2024-11-25
> > **Re: rebuttal**
> >
> > Thank you for the in-depth explanation. The authors' response was not only clear and detailed but also showcased a good understanding of their research. They have made commendable efforts to incorporate the feedback received. However, with the inclusion of the register token and guidance loss, I find that the approach's novelty is somewhat constrained. While these additions are indeed beneficial, they do not significantly enhance the originality. Furthermore, the comparison with BOQ-SA, which also utilizes slot attention, indicates only marginal improvements. Accordingly, I maintain my original scoring.

---

> > > ### Author Response · Authors · 2024-11-26
> > > **Official comment by authors**
> > >
> > > We thank the reviewer for their thoughtful comments.
> > >
> > > First, we would like to address a potential misunderstanding. The purpose of the additional BO-QSA experiment was not to further improve our results. Our results already surpass the current state-of-the-art by a significant margin in both segmentation and compositional generation. In our original submission, we already provided quantitative evaluations for segmentation performance, while reconstruction and compositional generation results were presented qualitatively. Following the reviewers’ suggestions, we conducted additional experiments during the rebuttal process to quantitatively evaluate our compositional generation results. These new evaluations demonstrate that our compositional generation significantly outperforms baseline methods. Please refer to the updated PDF file, which we have recently uploaded, for these results.
> > >
> > > The additional BO-QSA experiment specifically stemmed from the reviewer Hqxz’s question about the dependency of our compositional generation method on the segmentation step (to which the answer is clearly affirmative). Since our aim is not merely segmentation but to derive object-centric representations (each attending to a segment in the image) for generation purposes, we evaluated BO-QSA—a state-of-the-art extension of the original slot attention method. In our setup, BO-QSA yielded only a marginal improvement in segmentation performance. This reinforces the view that our segmentation results were already very competitive, rather than indicating any inherent weakness. We hope this clarification positively informs the reviewer’s interpretation of the BO-QSA experiment’s outcome.
> > >
> > > Regarding the novel contributions of our work, we emphasize that our use of adapters is substantially different from their use in other diffusion models. In our approach, the input to the adapter layers is not some fixed auxiliary data (e.g., depth or sketches), but rather the output of the slot attention network, which itself is learned during training. To the best of our knowledge, this is also the first time adapters have been employed in the context of object-centric learning (OCL). Furthermore, by introducing the register token and the guidance loss, our method achieves the best segmentation and generation results reported to date.
> > >
> > > In addition to these methodological novelties, a key contribution of our paper is its findings, which highlight the potential of OCL to enable plausible compositional generation and editing of real images. As noted by reviewer qbBW, this represents a significant milestone in the field.

---

> > > > ### Author Response · Authors · 2024-11-29
> > > > **Official Comment by Authors**
> > > >
> > > > We thank the reviewer for their positive assessment and constructive suggestions to strengthen the paper. With the rebuttal period nearing its deadline and the reviews being finalized, we want to assure the reviewer that we are ready to address any further concerns or questions that the reviewer may have.
> > > >
> > > > Specifically, we would be happy to further clarify the outcome of the additional BO-QSA experiment, which the reviewer had interpreted negatively. Although we addressed this issue in our previous response, we are ready to provide further explanations should the reviewer have any additional questions.

---

> ### Author Response · Authors · 2024-11-20
> **[accidently posted the same review twice]**
>
> accidentally posted the same response twice, response deleted

---

### Official Review · Reviewer_uBXM · 2024-11-06

**Soundness:** 3
**Presentation:** 3
**Contribution:** 2
**Rating:** 6
**Confidence:** 3

**Summary:**

This paper aims to incorporate object-centric information into the pre-trained diffusion models. Specifically, they argue that previous works have struggled with text-conditioning bias even though they can leverage the potential of pre-trained diffusion models. To eliminate the text-conditioning-related bias, the authors propose SlotAdapt, which utilizes the adapter layers for the condition with slot, to enable the incorporation of diverse slot conditions. For their design choice, T2I adapter design and register tokens-like approaches are introduced. Then, they also use cross-attention masks for better conditioning, which enables a dual path of guidance for the diffusion models and masks. Through their experiments, they present various benchmarks and comparisons with previous object-centric methods to validate the effectiveness of their method.

**Strengths:**

- The major strength is the performance of the proposed method. In their benchmark, their method significantly outperforms the baseline.
- The paper is well-written and easy to follow.

**Weaknesses:**

- W1: A significant concern with the proposed method is its novelty. While the object-centric approach may offer a fresh perspective within the realm of object-centric learning, the use of adapter layers is a well-established technique in diffusion fields, widely known to be effective. This familiarity may make the method appear somewhat trivial or lackluster in terms of innovation for diffusion model applications.

- W2: Additionally, the paper lacks an ablation study, which is critical for understanding the impact of the guidance loss on the model’s performance. Specifically, it would be insightful to see how the model behaves without the guidance loss, as this could further clarify the contribution of each component in the proposed approach.

**Questions:**

Please see the weakness.

---

> ### Author Response · Authors · 2024-11-20
> **Response to Reviewer uBXM**
>
> We thank reviewer uBXM for their detailed comments and feedback. They highlighted our work's strong performance and noted that the paper is well-written and easy to follow. However, they expressed concerns about the method's novelty. While we acknowledge that adapters are widely used in diffusion literature, our work is the first to apply them to the object-centric learning problem with a distinct motivation: utilizing information from the pretrained diffusion model without disrupting its prior knowledge. Moreover, our method introduces two additional key contributions: the register token and guidance loss.
>
> The reviewer's second point addressed the lack of ablation studies regarding the effects of guidance loss. We note that Table 2 in the main paper already presents these results, demonstrating the impact of different guidance loss types. The data shows that implementing guidance loss improves performance across all metrics except FG-ARI.

---

> > ### Comment · Reviewer_uBXM · 2024-11-25
> >
> > Thank you for your response. My concerns are addressed.

---

> > > ### Author Response · Authors · 2024-11-29
> > > **Official Comment by Authors**
> > >
> > > We thank the reviewer for their positive assessment and constructive suggestions to strengthen the paper. With the rebuttal period nearing its deadline and the reviews being finalized, we want to assure the reviewer that we are ready to address any further concerns or questions that the reviewer may have.

---

### Author Response · Authors · 2024-11-20
**Response for all reviewers**

We are grateful to all reviewers for their thorough comments and valuable feedback. The reviewers highlighted several strengths of our work, including impressive segmentation performance (uBXM, Hqxz), strong compositional generation capabilities (Hqxz, qbBW), well-defined motivation (Hqxz), and the significance of the problem we address (8w5Y). They also acknowledged our novel and effective contributions in design (qbBW, Hqxz). Despite noting some grammatical errors (Hqxz, qbBW), the reviewers found the paper well-written and easy to follow (uBXM, 8w5Y).

Reviewer uBXM expressed concerns about novelty, noting that adapter design is widely used in diffusion models. While we acknowledge this observation, our work represents the first application of adapters to the object discovery task, supported by strong theoretical motivation. Additionally, we introduce a novel guidance loss that improves performance across nearly all metrics (Table 2 in the main paper).

Reviewer Hqxz raised similar concerns about the increasing prevalence of adapters in LDMs. As mentioned above, our work's contribution extends beyond adapters to include a novel guidance loss. They also questioned whether our paper's outcomes depend on segmentation performance and inquired about the potential impact of improved segmentation methods. We clarify that we use slot attention for learning representations rather than purely for segmentation. To address this point, we conducted experiments replacing slot attention with its enhanced version, BOQ-SA, to demonstrate improvements.

Reviewer 8w5Y questioned our evaluation of reconstruction and compositional generation capabilities. Our additional experiments (detailed in our response to reviewer 8w5Y) demonstrate that our method outperforms baselines in both tasks. They also noted a potential limitation: using pretrained diffusion models might underperform on synthetic datasets like CLEVR/CLEVRTex. We acknowledge this limitation, as unlike previous work, we do not train the Diffusion model and VAE. The reviewer suggested including additional relevant references, which we will incorporate in the revised paper. Regarding their questions about guidance losses and potential seed-related variations, we explain in our response that the differences are not seed-dependent and that the guidance losses leverage the generative prior in the pretrained diffusion model.

Reviewer qbBW identified several grammatical errors and missing references, which will be addressed in the revised version. They also posed questions about the guidance loss, model differences, v-prediction, CFG scale, and suggested a new approach for register tokens. We have conducted additional experiments to address all these questions comprehensively (see response to qbBW).

---

### Author Response · Authors · 2024-11-25
**Updated version of the paper**

Dear Reviewers and Area Chairs,

Thank you for your thorough reviews and valuable feedback. We have carefully revised our submission to address your comments and suggestions. The updated PDF now includes the following enhancements, with all new additions marked in blue:

In the main paper:

- Expanded the related work section with the recommended citations
- Refined our terminology, replacing "near-perfect" with more precise descriptions of image fidelity
- Enhanced the limitations section to address synthetic data adaptation challenges

In the appendices:

- Section A.4: Added comprehensive experimental results on:
     - Better segmentation model (BOQ-SA)
     - Impact of varying slot numbers
     - Effects of additional slot as a register token implementation
- Section A.5: Incorporated quantitative evaluations using:
     -  FID (Fréchet Inception Distance) metrics and KID (Kernel Inception Distance) scores for both reconstruction and compositional generation tasks
- Section A.6: Extended visual demonstrations including:
    - Generations with Classifier-Free Guidance (CFG) at 1.3
    - Comparative analysis across different CFG values
    -  Seed variation studies

We appreciate your constructive feedback that has helped strengthen our paper.

Best regards,

---

### Meta-Review · Area_Chair_i4bE · 2024-12-17

**Metareview:**

This paper introduces slot-based conditioning in generative diffusion image models. The approach builds on existing pre-trained text-conditioned diffusion models and adds an adapter to slot-conditioning. In addition a novel guidance loss is introduced to improve performance. The proposed model improves over baselines in object discovery and compositional generation.
The strengths mentioned in the reviews include: paper well written, performance gains over baselines, and pushing the boundaries of object-centric learning to real-world images,
Weaknesses: novelty wrt generative diffusion literature, lack of comparison to several baselines, overclaims in different places, missing discussion of some related work, performance on simpler datasets like CLEVR not reported.

**Additional Comments On Reviewer Discussion:**

In response to the reviews the authors submitted a rebuttal and a revised version of the manuscript. The rebuttal addressed most of the points raised in the reviews, and after taking it into account as well as the other reviews, three of four reviewers recommend accepting the paper. The AC does not see any major concerns from the remaining reviewer to overturn the majority recommendation.

---

### Decision · Program_Chairs · 2025-01-22

Accept (Poster)